# Structure of universal formulas

**Dmitry Yarotsky**
Skoltech
d.yarotsky@skoltech.ru

## Abstract

By universal formulas we understand parameterized analytic expressions that have a fixed complexity, but nevertheless can approximate any continuous function on a compact set. There exist various examples of such formulas, including some in the form of neural networks. In this paper we analyze the essential structural elements of these highly expressive models. We introduce a hierarchy of expressiveness classes connecting the global approximability property to the weaker property of infinite VC dimension, and prove a series of classification results for several increasingly complex functional families. In particular, we introduce a general family of polynomially-exponentially-algebraic functions that, as we prove, is subject to polynomial constraints. As a consequence, we show that fixed-size neural networks with not more than one layer of neurons having transcendental activations (e.g., sine or standard sigmoid) cannot in general approximate functions on arbitrary finite sets. On the other hand, we give examples of functional families, including two-hidden-layer neural networks, that approximate functions on arbitrary finite sets, but fail to do that on the whole domain of definition.

## 1 Introduction

By *universal formulas* we broadly (informally) understand parameterized explicit analytic expressions that have a fixed complexity (as expressions) but nevertheless can approximate any continuous function on a compact set. An example of such formula, given by Boshernitzan [1], is

$$y(x) = \int_0^{x+a} \frac{bd}{1 + d^2 - \cos(bt)} \cos(e^t) dt + c, \tag{1}$$

where $d > 0$, $a, b$ and $c$ are parameters. [1] proves that, by varying the parameters, functions (1) can uniformly approximate any continuous function on any compact interval in $\mathbb{R}$.

Expression (1) involves integration. Laczkovich and Ruzsa [13] prove existence of universal formulas representable as elementary functions without integration (moreover, their formulas can uniformly approximate continuous functions on the whole $\mathbb{R}$ under a mild growth assumption).

One can further restrict the types of operations and show that universal formulas can be realized by fixed-size classical neural networks with suitable activation functions [29]. By a classical neural network we mean a computational model consisting of units ("hidden neurons"), each computing a map $z_1, \ldots, z_n \mapsto \sigma(\sum_{k=1}^n w_k z_k + h)$, where $z_1, \ldots, z_n$ (signals) are outputs of some previous neurons, and $w_k$ and $h$ are weights (parameters) of the neuron. In addition to hidden neurons, one distinguishes "input neurons" that just represent the scalar components of the network input, and "output neuron(s)" that represent the scalar component(s) of the output. The output neurons work as hidden neurons but without activations, i.e. as $z_1, \ldots, z_n \mapsto \sum_{k=1}^n w_k z_k + h$. As defined, neural networks are fairly restrictive in terms of used operations: for example, they contain multiplications of signals $z$ by constants (weights) $w$, but not multiplications between signals, $z_1 z_2$. However, this and some other extra operations can be implemented with any accuracy using suitable combinations of neurons. The universal network in [29] uses a fixed architecture graph, activations $\sin$ and $\arcsin$,

37th Conference on Neural Information Processing Systems (NeurIPS 2023).

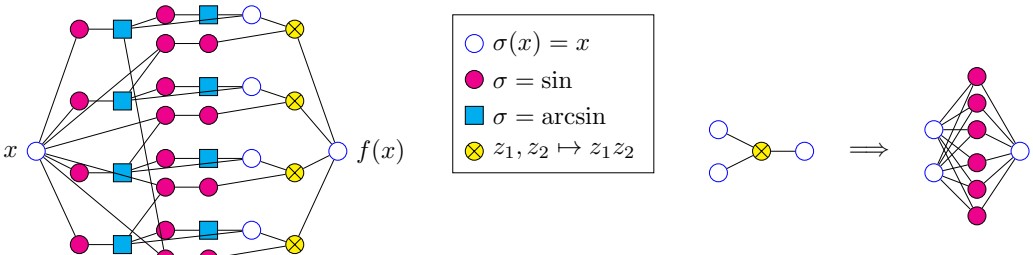

Figure 1: **Left:** A (slightly simplified) fixed-size universal neural network approximating any $f \in C([0, 1])$ from [29]. Most hidden neurons are standard neurons $z_1, \ldots, z_n \mapsto \sigma(\sum_{k=1}^{n} w_k z_k + h)$ with one of the activation functions $\sigma$ (identity, sin or arcsin). Also, there are several multiplication neurons. The parameters of the model are all the weights $w_k, h$ in the standard neurons. **Right:** A multiplication neuron can be replaced by a group of standard neurons while preserving universality.

and can approximate any $f \in C([0, 1])$ (see Figure 1). Another fixed-size universal network had been constructed earlier in [15], but the activation was not an elementary function in that work.

The Kolmogorov(-Arnold) Superposition Theorem [10] shows that any function $f \in C([0, 1]^d)$ can be expressed in terms of a fixed number (only depending on $d$) of summations and univariate continuous functions. This implies that multivariate universal formulas can be easily constructed from univariate ones, by plugging the latter in the Kolmogorov ansatz. Accordingly, the question of multivariate universal formulas is not significantly more interesting or complicated than its counterpart for univariate formulas, and in this paper restrict our attention to the latter. We also mention in passing another interesting property of universal formulas: they naturally give rise to universal polynomial differential equations $P(x, f, f', \ldots, f^{(n)}) = 0$, with a polynomial $P$ and a dense set of solutions in $C([a, b])$, see [21, 2, 1, 19, 13]. (In fact, one can find such a polynomial for any parameterized family of elementary functions; the density of solutions then follows from the universality of the formula.) See [22] for a connection to analog computers.

A foundational theorem on neural networks is the universal approximation theorem (UAT). It states that, for a broad class of activation functions, a neural network with a single layer of hidden neurons can approximate any continuous function on a compact set if we increase the number of neurons [4, 18]. This result is very different from, and should not be confused with the above *fixed-size* universal networks of [15] or [29]. In UAT, universality is achieved by increasing the number of parameters, and accordingly UAT imposes much weaker requirements on the activation functions (e.g., it holds for any continuous non-polynomial activation, see [14, 18]).

On the other hand, the fixed-size universal network in Figure 1 is fairly simple. One of the two activation functions used there, sin, is actually used as an activation in practical multi-layer networks (e.g., the popular SIREN network of [23]). This raises the following natural question motivating the present paper: *can practically used neural-network-type models be universal as fixed-size formulas (i.e., without increasing the number of neurons)?*

There is an important caveat here: the universality of universal formulas is, of course, an idealization that requires their parameters to have unbounded precision or magnitude. If parameters are finitely defined so as to be representable on real computer, one can give a simple entropy bound constraining feasible approximation accuracy. Specifically, suppose that a formula contains $p$ parameters, each somehow represented using $B$ bits (e.g., $B = 32$ or 64 for standard floats). Suppose we are approximating a function $f$ at $N$ points $x_1, \ldots, x_N$ with absolute accuracy $\epsilon$. Suppose finally that $|f|$ is bounded by $M$ and $f(x_k), k = 1, \ldots, N$, can be chosen independently subject to this constraint. Then

$$N \log_2(M/\epsilon) \le pB. \tag{2}$$

This shows that in a practical application with parameters implemented as standard floats, universal formulas such as (1) can typically produce only crude approximations. In the sequel, we will ignore these finite-precision aspects.

**Our contribution.** Our goal in this paper is to understand which structural properties are critical for universal formulas. This question does not seem to be simple. As a general strategy of tackling it, we analyze a sequence of progressively more complex parametric functional families with respect to several relevant classes of model expressiveness. Our specific contributions are as follows.

1. We introduce a hierarchy $\mathcal{G}_0 \supset \mathcal{G}_1 \supset \mathcal{G}_2$ of classes of model expressiveness (Section 2). Here $\mathcal{G}_0$ represents the families of infinite VC-dimension, $\mathcal{G}_1$ the families achieving approximation on arbitrary finite sets, and $\mathcal{G}_2$ the families achieving uniform approximation on the whole segment $[0, 1]$. Universal formulas correspond to the class $\mathcal{G}_2$. The class $\mathcal{G}_0$ is important because elementary functions belonging to it must involve, in some form, the function $\sin$ acting on an unbounded domain (Section 3). We argue that the class $\mathcal{G}_1$ is also important because, as we show, some functional families cannot belong to it due to algebraic (polynomial) constraints.

2. As a first result on algebraic constraints, we consider the family of fixed-size linear combinations of sine waves with unbounded weights (equivalently, fixed-size single-hidden-layer neural networks with activation $\sin$) and prove that it belongs to the class $\mathcal{G}_0 \setminus \mathcal{G}_1$. Moreover, we give a complete description of the limit points of this family (Section 4).

3. As a generalization of the previous result, we introduce a general family of *polynomially-exponentially-algebraic* expressions of bounded complexity, and prove that it is also subject to polynomial constraints and lies outside $\mathcal{G}_1$ (Section 5).

4. We give an application of the previous result to branching expressions and neural networks with conventional activations (Section 6). Specifically, we prove that if the network has a bounded complexity and possibly multiple layers, but only one layer with transcendental activation functions (such as $\sin x$, Gaussian or standard sigmoid), then the network is subject to polynomial constraints and lies outside $\mathcal{G}_1$. Moreover, polynomial constraints hold even if the activation functions are only *piecewise* analytic, as is common in practice.

5. The situation is drastically different for formulas involving compositions of non-polynomial analytic functions and $\sin$: such formulas generally belong to $\mathcal{G}_1$ (Section 7). It seems hard to separate the classes $\mathcal{G}_1 \setminus \mathcal{G}_2$ and $\mathcal{G}_2$. We give a couple of examples of families for which we can prove that they belong to $\mathcal{G}_1 \setminus \mathcal{G}_2$. One of these families can be described as that of fixed-complexity two-hidden-layer networks with $\sin$ activation that have a single neuron in the second hidden layer. We conjecture that general sine networks of fixed complexity also belong to $\mathcal{G}_1 \setminus \mathcal{G}_2$.

Most proofs are deferred to the appendix (the respective sections are indicated in the theorems).

## 2 The hierarchy of expressiveness classes

Throughout the paper, we consider various families $H$ of functions $g : [0, 1] \to \mathbb{R}$. We define for them expressiveness classes $\mathcal{G}_0, \mathcal{G}_1, \mathcal{G}_2$:

$H \in \mathcal{G}_0 \overset{\text{def}}{\Longleftrightarrow}$ Vapnik-Chervonenkis dimension $\mathrm{VCdim}(H) = \infty$;

$H \in \mathcal{G}_1 \overset{\text{def}}{\Longleftrightarrow}$ For any *finite* subset $\{x_1, \ldots, x_N\} \in [0, 1]$, any $y_1, \ldots, y_N \in \mathbb{R}$ and any $\epsilon > 0$ there is $g \in H$ such that $|g(x_k) - y_k| < \epsilon$ for all $k = 1, \ldots, N$.

$H \in \mathcal{G}_2 \overset{\text{def}}{\Longleftrightarrow}$ For any $f \in C([0, 1])$ and $\epsilon > 0$, there is $g \in H$ such that $\|g - f\|_\infty = \sup_{x \in [0,1]} |g(x) - f(x)| < \epsilon$.

Here, $\mathrm{VCdim}(H_f) = \infty$ means that for any $N \in \mathbb{N}$ we can find a size-$N$ subset $X \subset [0, 1]$ shattered by $H$ thresholded at 0, i.e. such that for any $S \subset X$ there is $g \in H$ for which $g(x) > 0$ if $x \in S$ and $g(x) \leq 0$ if $x \in X \setminus S$.

Clearly,

$$\mathcal{G}_0 \supsetneq \mathcal{G}_1 \supsetneq \mathcal{G}_2. \tag{3}$$

The classes $\mathcal{G}_1$ and $\mathcal{G}_2$ reflect approximability on arbitrary finite sets and on the whole interval $[0, 1]$, respectively. Universal formulas as discussed in the Introduction correspond to the class $\mathcal{G}_2$. One can

say that $\mathcal{G}_1$ corresponds to universality in a weaker sense, which is particularly relevant if we try to learn the parameters using a finite training set $\{(x_k, y_k)\}$.

We will consider several families $H$ that we will distinguish by superscripts denoting or enumerating their types (e.g. $H^f$ or $H^{(1)}, H^{(2)}, \ldots$) and by subscripts reflecting complexity within the type (e.g., $H_N^{(2)}$ will denote linear combinations of $N$ sine waves). Most of these families admit natural interpretations in terms of conventional neural networks – we will mention them along the way.

While the remainder of this work focuses on the mathematical analysis of the relation between various functional families and the classes $\mathcal{G}_k$, let us briefly clarify the role of these classes from the general perspective of approximation and learning.

1. Most conventional learning models (e.g., neural networks) can be said to be defined using (piecewise) elementary functions. If the model is assumed to have a bounded complexity (e.g., a bounded number of neurons) and its weights and signals are bounded, this model falls outside even the broadest class $\mathcal{G}_0$ (see Section 3 below). This does not apply, however, to models containing the function $\sin$ acting on an unbounded domain.

2. For a fixed-complexity model, being outside the classes $\mathcal{G}_k$ and thus not having the full approximation power is not a practically negative property: this just means that the complexity of the model needs to increase to achieve a better fit.

3. The situation when a fixed-complexity model belongs to the class $\mathcal{G}_2$ (i.e., is a universal formula) is mathematically interesting but somewhat exotic from the perspective of conventional computation, since the respective model parameters need to contain a potentially unbounded amount of information.

4. The class $\mathcal{G}_1 \setminus \mathcal{G}_2$ seems especially dangerous from the generalization point of view: given a hypothesis space $H$ from this class, we can fit a model arbitrarily well to a generic continuous ground truth on any finite training set, but can hardly ever uniformly approximate this ground truth on the whole domain. This cannot be said about the class $\mathcal{G}_2$. In Section 7 we give examples of functional families (neural networks) provably belonging to this dangerous class $\mathcal{G}_1 \setminus \mathcal{G}_2$.

## 3 Pfaffian functions

There is an important broad class of parameterized functions $f(x, \mathbf{w})$ for which one can guarantee that the family $H^f$ obtained by fixing various $\mathbf{w}$ has a finite VC-dimension. This class is most naturally described in terms of *Pfaffian* functions introduced by [9]. Pfaffian functions can be defined using function sequences in which, at each step, the next function satisfies polynomial first-order differential equations involving itself and previous functions (see [9, 5, 30] for details). In particular, Pfaffian functions include all elementary functions when the latter are defined on suitable domains.

Importantly, in contrast to functions like $\ln, \exp$ and polynomials, which are Pfaffian on their whole natural domain of definition, the function $\sin$ is not Pfaffian on the whole set $\mathbb{R}$, though it is Pfaffian on any bounded interval $(a, b)$ (moreover, the representation of $\sin$ as a Pfaffian function depends on $(a, b)$ and becomes more complex as $b - a$ increases). For this reason, elementary functions are Pfaffian only when they involve $\sin$ (or a closely related function, like $\cos$) restricted to bounded intervals.

The fundamental theorem of [9] gives an explicit finite upper bound on the number of zeros of Pfaffian functions, which implies, in particular, that the respective parametric families have finite VC-dimension. We state this result specialized to elementary functions and in the form suitable for our purposes; see [9, 5, 30, 29] for details.

**Theorem 1.** *Suppose that a formula $y = f(x, \mathbf{w})$ is constructed from the variables $x$ and $w_1, \ldots w_d$ using real numbers, standard arithmetic operations $(+, -, \times, /)$, elementary functions $\ln$, $\exp$, $\sin$, $\arcsin$, and compositions. Suppose that there is a nonempty subset $W \subset \mathbb{R}^d$ such that for any $\mathbf{w} = (w_1, \ldots, w_d) \in W$ the function $f(\cdot, \mathbf{w}) : [0, 1] \to \mathbb{R}$) is well-defined, i.e., $\ln$ and $\arcsin$ are applied on the intervals $(0, \infty)$ and $(-1, 1)$, respectively, and there is no division by 0. Moreover, suppose that there is a bounded interval $(a, b)$ to which the arguments of $\sin$ always belong for all $\mathbf{w} \in W$. Then the family $H^f = \{f(\cdot, \mathbf{w})\}_{\mathbf{w} \in W}$ has a finite VC-dimension.*

This theorem shows that an elementary function has chances to be universal only if it includes $\sin$ (or some related function) acting on an unbounded domain. Therefore, we will focus on these latter functions in the sequel.

It is crucial for Theorem 1 that all the computations in the formula are real-valued and not complex-valued (the proof eventually boils down to Rolle's theorem, which does not hold for $\mathbb{C}$-valued functions). In particular, we cannot bypass the domain restriction of $\sin$ by simply writing $\sin x = (e^{ix} - e^{-ix})/2i$. In contrast, many results in the next sections will be essentially algebraic and will hold for complex-valued operations.

We remark that Sontag [25] considers a class of formulas closely related to Pfaffian functions and proves that shattering $k$-point sets in general position requires at least $(k-1)/2$ parameters in such formulas.

## 4 Linear combinations of sines

A simple example of a family with an infinite VC-dimension is $H^{(1)} = \{c\sin(\omega x)\}_{c,\omega\in\mathbb{R}}$. This is usually proved (see e.g. [28], Section 3.6) by giving a particular example of an arbitrarily large finite set shattered by $H^{(1)}$. For our purposes, however, it is instructive to describe a general collection of finite sets on which $H^{(1)}$ has universal approximability. We say that numbers $x_1, x_2, \ldots$ are *rationally independent* if they are linearly independent over the field $\mathbb{Q}$ of rational numbers.

**Theorem 2.** *Let* $X = \{x_1, \ldots, x_N\}$ *be a finite subset of* $[0,1]$ *such that the values* $x_1, \ldots, x_N$ *are rationally independent. Then the family* $H^{(1)}$ *can approximate any* $\mathbb{R}$*-valued function on* $X$.

*Proof.* Consider the torus $\mathbb{T}^N = (\mathbb{R}/2\pi\mathbb{Z})^N$; its points can be identified with the points of the cube $[0, 2\pi)^N$. Let $\lfloor\cdot\rfloor : \mathbb{R} \to [0,1)$ be the standard floor function. By a classical Kronecker's theorem [11, 7], the set $\{(2\pi(\theta x_1 - \lfloor\theta x_1\rfloor), \ldots, 2\pi(\theta x_N - \lfloor\theta x_N\rfloor))\}_{\theta\in\mathbb{R}}$ is a dense subset of $\mathbb{T}^N$ if and only if the numbers $x_1, \ldots, x_N$ are rationally independent. Using the $2\pi$-periodicity of $\sin$, this implies immediately that if $x_1, \ldots, x_N$ are rationally independent, then $\{(\sin(\omega x_1), \ldots, \sin(\omega x_N))\}_{\omega\in\mathbb{R}}$ is dense in $(-1, 1)^N$ and hence $\{(c\sin(\omega x_1), \ldots, c\sin(\omega x_N))\}_{c,\omega\in\mathbb{R}}$ is dense in $\mathbb{R}^N$. $\square$

This result shows that the family $H^{(1)} \in \mathcal{G}_0$. It is also clear that $H^{(1)} \notin \mathcal{G}_1$ (e.g., since $g(0) = 0$ for any $g \in H^{(1)}$). In fact, the sufficient condition of rational independence in the theorem is close to also being necessary. In particular, a simple kind of sets not shattered by $H^{(1)}$ are all nontrivial arithmetic progressions $x_k = u + vk, v \neq 0$, of length 5. Indeed, it follows from the identity $\sin(u + (k-1)v) + \sin(u + (k+1)v) = 2\sin(u + kv)\cos(v)$ that the values of functions from $H^{(1)}$ cannot, for example, have the sign configuration $+ + + - +$ on such progressions.

Consider now the more general family consisting of all linear combinations of a fixed number of sine waves with arbitrary frequencies and phases:

$$H_N^{(2)} = \left\{\sum_{n=1}^{N} c_n \sin(\omega_n x + h_n)\right\}_{c_n,\omega_n,h_n\in\mathbb{R},n=1,\ldots,N}. \tag{4}$$

One can view this family as representing single-hidden-layer neural networks with the $\sin$ activation function ($\omega_n$ and $h_n$ are the weights and biases in the hidden layer, and $c_n$ are the weights in the output layer).

One can again easily show that $H_N^{(2)} \in \mathcal{G}_0 \setminus \mathcal{G}_1$, by observing that the values of all functions $g \in H_N^{(2)}$ are constrained by common nontrivial polynomial equations:

**Theorem 3.**

1. *For any* $g \in H_N^{(2)}$, $x_0 \in \mathbb{R}$ *and* $\alpha_n, \beta_n \in \mathbb{R}, n = 0, \ldots, 2N$,

$$\det A = 0, \text{ where } A = \left(g(x_0 + \alpha_k + \beta_m)\right)_{k,m=0}^{2N}. \tag{5}$$

2. $\det A$ *is a nonzero polynomial in the variables* $g(x)$, *where* $x \in X = \{x_0 + \alpha_k + \beta_m | k, m = 0, \ldots, 2N\}$, *if and only if all* $\alpha_k$ *are different and all* $\beta_m$ *are different.*

*Proof.* 1. The matrix $A$ is degenerate because each of its $2N + 1$ rows is a linear combination of $2N$ vectors $(e^{s\omega_n \beta_1}, \ldots, e^{s\omega_n \beta_N})$ with $n = 1, \ldots, N$ and $s = \pm i$.

2. If some $\alpha_k$ or $\beta_m$ are repeated, then the matrix $A$ contains repeated columns or rows and so the polynomial $\det A$ identically vanishes. Conversely, suppose that all $\alpha_k$ are different and all $\beta_m$ are different. Without loss, we can assume $\alpha_0 < \ldots < \alpha_{2N}$ and $\beta_0 < \ldots < \beta_{2N}$. We can then expand $\det A = g(x_0 + \alpha_{2N} + \beta_{2N}) \det A' + \ldots$, where $\det A'$ is the top left $2N \times 2N$ minor and the terms $\ldots$ do not contain the variable $g(x_0 + \alpha_{2N} + \beta_{2N})$. Repeating the expansion for $A'$, etc., we see that the polynomial $\det A$ contains the monomial $\prod_{k=0}^{2N} g(x_0 + \alpha_k + \beta_k)$ with coefficient 1, i.e. is nonzero. $\qquad\square$

This result shows that there are (infinitely many) nontrivial polynomial constraints on the values of the functions $g \in H_N^{(2)}$. Importantly, these constraints are preserved under pointwise limits, so, in particular, $H_N^{(2)} \in \mathcal{G}_0 \setminus \mathcal{G}_1$. Taking $N = 1$ and $\alpha_k = \beta_k = vk, k = 0, 1, 2$, we obtain a nontrivial constraint for the values of $g$ on any nontrivial length-5 arithmetic progression $x_k = u + vk$, in agreement with an earlier observation.

In fact, thanks to the semi-linear structure of the family $H_N^{(2)}$, we can go further and give an explicit description of its limit points. It turns out that the limiting functions can only differ from the original combinations of sine waves by resonant factors $x^m$:

**Theorem 4** (A). *Let $N$ be fixed. Then a function $f_*$ is a limit point of $H_N^{(2)}$ in the sense of uniform convergence on the segment $[0, 1]$ if and only if*

$$f_*(x) = \sum_{m=0}^{2M_0 - 1} c_{0m} x^m + \sum_{k=1}^{K} \sum_{m=0}^{M_k - 1} c_{km} x^m \sin(\omega_k x + h_{km}), \tag{6}$$

*where $M_0 \geq 0$, $M_k \geq 1$ for $k = 1, \ldots, K$, and $\sum_{k=0}^{K} M_k = N$.*

Resonant factors appear by taking suitable combinations of sine waves at close frequencies, e.g. $x \sin(\omega x) = \lim_{\Delta x \to 0} (\Delta \omega)^{-1} (\sin((\omega + \Delta \omega)x) - \sin(\omega x))$. We show explicitly how to construct resonances of arbitrary degrees subject to the constraint $\sum_{k=0}^{K} M_k = N$. Note that the frequency $\omega = 0$ corresponding to the first sum in expansion (6) is special as it is associated with a potential double-degree resonance.

The more difficult part of the proof is the "only if" part, i.e. that any limiting function has the form (6). Our basic idea is to show that a limiting function is subject to a linear differential equation describing waves with a given set of frequencies; the general resonant form (6) would then appear from the usual solution of such an equation with nontrivial multiplicities of the frequencies. This strategy, however, is not so easy to implement rigorously, since in our setting the frequencies are unbounded. We give two alternative proofs overcoming this difficulty. The first, *clustering-based* proof, groups the frequencies found in a converging sequence $f_n \in H_N^{(2)}$ into spatially separated "frequency clusters", and then shows that unbounded clusters can be removed without changing the limit. The second, *discretization-based* proof, starts with describing the discretized functions of $H_N^{(2)}$ by suitable linear finite-difference equations which also hold for the limiting function $f_*$. Then, we consider these discretizations at different length scales and show that they can only be consistent with each other and with the continuity of the uniform limit $f_*$ if representation (6) holds.

## 5  Polynomially-exponentially-algebraic expressions

Theorem 3 shows that the values of functions $H_N^{(2)}$ are subject to nontrivial polynomial constraints preserved under pointwise limits. We extend this argument to a much larger family of functions, involving complex algebraic operations. This generaliztion will allow us, in particular, to prove a related result for multi-layer neural networks.

If $U \subset \mathbb{C}^N$ is an open domain, we say that a holomorphic function $Q : U \to \mathbb{C}$ is an *algebraic function* if there exist $N$-variable polynomials $\phi_0, \ldots, \phi_q$, not all zero, such that

$$\sum_{k=0}^{q} \phi_k(z_1, \ldots, z_N) Q^k(z_1, \ldots, z_N) = 0, \quad \forall \mathbf{z} = (z_1, \ldots, z_N) \in U. \tag{7}$$

The value $q$ is called the *degree* of the algebraic function $Q$. Algebraic functions include, in particular, all polynomials and (on their holomorphy domains) rational functions and powers $z^a$ with rational $a$. Arithmetic operations and compositions applied to algebraic function produce again algebraic functions.

We define the family $H_{N,q,r,d}^{(3)}$ as consisting of all functions $g : [0,1] \to \mathbb{R}$ of the form

$$g(x) = Q(x, e^{P_1(x)}, \ldots, e^{P_N(x)}), \tag{8}$$

where $P_1, \ldots, P_N$ are complex-valued polynomials of degree $\leq d$, and $Q$ is an algebraic function of degree $\leq q$ such that all polynomials $\phi_k$ appearing in its definition have degree $\leq r$. Here, the functions $Q$ are assumed to be defined on some domains $U \subset \mathbb{C}^{N+1}$ that contain all the values $(x, e^{P_1(x)}, \ldots, e^{P_N(x)}), x \in [0,1]$. Our main result is the following extension of Theorem 3.

**Theorem 5** (B). *Fix $N, q, r, d$. Let $m = (q+1)\binom{N+r+1}{N+1} + (\max(1,d)+1)(N+1)$. Then there exists a nonzero polynomial $R$ of $m+1$ variables such that for any $g \in H_{N,q,r,d}^{(3)}$ and any $a, b \in [0,1]$*

$$R(g(a), g(a+h), g(a+2h), \ldots, g(b)) = 0, \quad h = \tfrac{b-a}{m}. \tag{9}$$

As a corollary, since the polynomial $R$ is the same for all $g \in H_{N,q,r,d}^{(3)}$, constraint (9) is preserved by pointwise limits, so $H_{N,q,r,d}^{(3)} \notin \mathcal{G}_1$.

Our proof of Theorem 5 is inspired by the theory of algebraic-transcendent functions [16, 17, 8] that addresses representability of functions by algebraic differential equations. Our context is rather different: we are interested in the pointwise convergence of functions (or convergence in the uniform norm), so conditions involving derivatives are not applicable in our setting. We observe, however, that some elements of this theory can be extended from the usual to discretized derivatives. This is why, in particular, the conditions in Eq. (9) are formulated for a regular grid of points $a, a+h, \ldots, b$.

Since the polynomials $P_n$ in the definition of $H_{N,q,r,d}^{(3)}$ are allowed to be complex-valued, the families $H_{N,q,r,d}^{(3)}$ subsume the sine wave families $H_N^{(2)}$ considered in the previous section:

$$H_N^{(2)} \subset H_{2N,1,1,1}^{(3)}. \tag{10}$$

## 6 Branching expressions and neural networks

We describe now an application of Theorem 5 to neural networks. We start by noting that many standard activation functions can be classified as piecewise polynomial or (piecewise) algebraic-exponential[1]:

**Piecewise polynomial:** Binary step function $\sigma(x) = \mathbf{1}[x \geq 0]$, ReLU $\sigma(x) = \max(0,x)$, leaky ReLU $\sigma(x) = \max(ax, x)$, squared ReLU $\sigma(x) = \max^2(0,x)$ [24].

**(Piecewise) algebraic-exponential:** $\sin x$, $\tanh x$, standard sigmoid $\sigma(x) = (1 + e^{-x})^{-1}$, ELU

$$\sigma(x) = \begin{cases} a(e^x - 1), & x < 0 \\ x, & x \geq 0 \end{cases} \text{ [3], swish } \sigma(x) = x/(1 + e^{-x}) \text{ [20], Gaussian } \sigma(x) = e^{-x^2}.$$

"Piecewise" here means that $\mathbb{R}$ is in general divided into several subsets (in fact, up to two subsets in the above examples) on which the activation is defined by different analytic formulas. We will refer to this as "branching". On each branch, activations of the first type are polynomials, and those of the second type can be represented in the form (8) with a polynomial or rational $Q$.

Define the family $H_N^{(4)}$ of functions $g : [0,1] \to \mathbb{R}$ as realization, with various weight assignment, of the following class of feedforward neural networks. Suppose that the underlying directed acyclic graph of the network contains at most $N$ neurons. Suppose that each hidden neuron is equipped with one of the piecewise polynomial or algebraic-exponential activation functions mentioned above (possibly different at different neurons). Suppose finally that any directed path from the input to the output neuron contains *at most one* (piecewise) algebraic-exponential neuron (for example, this is the case if the network has a single algebraic-exponential layer, while the other layers are piecewise polynomial). Then we have

---

[1]Of course, there also exist standard activations not of these two types, e.g. softplus $\sigma(x) = \ln(1 + e^x)$ [6].

**Theorem 6** (C). *Given $N$, there exists $m$ and a nontrivial polynomial $R$ of $m + 1$ variables such that for any $g \in H_N^{(4)}$ and any $a, b \in [0, 1]$*

$$R(g(a), g(a + h), g(a + 2h), \ldots, g(b)) = 0, \quad h = \tfrac{b-a}{m}. \tag{11}$$

The main obstacle in deriving this theorem from Theorem 5 is the branching in the activations that destroys the global analytic structure of functions $g \in H_N^{(4)}$. We overcome this obstacle using the well-known Van der Waerden's theorem on arithmetic progressions.

**Theorem 7** ([27]; see, e.g., [26] for a short proof). *For any integers $s, p \geq 1$ there exists an integer $N_{vdW}(s, p) \geq 1$ such that every coloring $C : \{1, ..., N_{vdW}\} \to \{1, ..., p\}$ of $\{1, ..., N_{vdW}\}$ into $p$ colors contains at least one monochromatic arithmetic progression of length $s$ (i.e. a progression in $\{1, ..., N_{vdW}\}$ of cardinality $s$ on which $C$ is constant).*

## 7    Compositions of sines and non-polynomial functions

The polynomially-exponentially-algebraic structure (8) is actually close to being necessary for the polynomial constraints of Theorem 5, as we show by the following example. Let $\sigma$ be a real analytic function on the interval $[-1, 1]$. Let $f(x) = c\sin(\omega\sigma(bx) + h)$ with parameters $c, b, h \in \mathbb{R}$ and $b \in [-1, 1]$. Denote the respective family by $H^\sigma$.

**Theorem 8** (D). *$H^\sigma \in \mathcal{G}_1$ if and only if $\sigma$ is not a polynomial.*

This shows that the polynomial constraints of Theorem 5 are destroyed if we relax even slightly the requirement of polynomial arguments of the exponentials in Eq. (8). They are also destroyed if we replace the algebraic function $Q$ by the sine function.

While $H^\sigma \in \mathcal{G}_1$ for non-polynomial $\sigma$, we can show that $H^\sigma$ is never in $\mathcal{G}_2$, by directly finding all limit points of $H^\sigma$ under uniform limits. Let us call the limit points of $H^\sigma$ not belonging to $H^\sigma$ *nontrivial*.

**Theorem 9** (E). *For a non-constant $\sigma$, the only nontrivial limit points of the family $H^\sigma$ are $a_0 + a_r x^r$, $a\sigma(bx) + c$, and $c\sin(a_0 + a_r x^r)$, where $r = \arg\min(m > 0 : \frac{d^m\sigma}{dx^m}(0) \neq 0)$ and $a_0, a_r, a, b, c \in \mathbb{R}$. In particular, $H^\sigma \notin \mathcal{G}_2$.*

The families $H^\sigma$ with non-polynomial $\sigma$ thus provide examples of elements of the class $\mathcal{G}_1 \setminus \mathcal{G}_2$ of finitely- but not globally-universal formulas. Consider now a more complex family $H_N^{(5)}$ consisting of the functions

$$f(x) = c\sin(g(x)) + h, \quad g \in H_N^{(2)}, \tag{12}$$

where $H_N^{(2)}$ consists of linear combinations of $N$ sine waves as defined earlier in Eq. (4). One can interpret the family $H_N^{(5)}$ as a two-hidden-layer neural network with the sin activation function, $N$ neurons in the first hidden layer, and a single neuron in the second hidden layer. Using the same approach of examining the limit points and invoking additional topological arguments, we prove that the family $H_N^{(5)}$ belongs to the same class as $H^\sigma$:

**Theorem 10** (F). *For any $N \geq 1$, $H_N^{(5)} \in \mathcal{G}_1 \setminus \mathcal{G}_2$.*

It appears that the class $\mathcal{G}_1 \setminus \mathcal{G}_2$ is quite generic – we conjecture that an analog of Theorem 10 holds for any multi-layer neural network with the sin activation and a bounded number of neurons. However, proving such a general result by explicitly describing the limit points as in Theorems 9, 10 is difficult. We are not aware of simple conditions allowing to separate $\mathcal{G}_1 \setminus \mathcal{G}_2$ from $\mathcal{G}_2$.

In this context, let us discuss the structure of universal formulas mentioned in the Introduction. Boshernitzan's formula (1) includes the factor $\frac{bd}{1+d^2-\cos(bt)}$ which is of the form (8) and hence in $\mathcal{G}_1$, and the transcendental composition $\cos(e^t)$. This agrees with our results showing that compositions of sines and non-polynomial functions are essential for universal formulas. However, Boshernitzan's formula also includes integration, which effectively serves to produce an approximation to a piecewise constant function. The neural network in Figure 1 and the formulas of [13] combine sines with arcsines, which serves to approximate periodic piecewise linear or piecewise constant functions. It seems that approximate piecewise constant functions are an essential element of existing universal

formulas, and so one can conjecture that arcsine or a related inverse trigonometric function is a necessary component of an elementary universal formula without integration.

In the fairly different context of recursive decidability, there is a remarkably similar sharp difference between polynomial-sine and more complex compositions [12]. Let $\mathcal{S}_1$ denote the ring generated by the integers and the expressions $x, \sin x^n$ and $\sin(x \sin x^n)(n = 1, 2, \ldots)$. Let also $\mathcal{S}_2$ denote the ring generated by the integers and the expressions $\sin x^n$ and $\cos x^n (n = 1, 2, \ldots)$. Then the predicate that there exists a real $x$ such that $f(x) > 0$ is undecidable for $f \in \mathcal{S}_1$, but decidable for $f \in \mathcal{S}_2$.

# 8   Discussion

**The expressiveness hierarchy.**   To analyze the necessary features of the structure of universal formulas, we have examined the chain of expressiveness classes $\mathcal{G}_0 \supset \mathcal{G}_1 \supset \mathcal{G}_2$ corresponding, respectively, to the families having the infinite VC-dimension, general approximability on all finite sets, and general global uniform approximability. Membership in the class $\mathcal{G}_0$ can be conveniently studied using the theory of Pfaffian functions, while, as we have shown, membership in the class $\mathcal{G}_1$ can be conveniently studied using algebraic methods. In particular, our results suggest that globally universal formulas need to involve suitable compositions of non-polynomial and sine operations.

**Transcendental structure of universal formulas.**   Our main result in Section 5 shows that combining a single layer of exponential and sine operations with polynomials and algebraic functions has a fairly weak expressiveness, in the sense that such formulas of bounded complexity are constrained by polynomial relations and do not even have general approximability on finite sets. In this family, the requirement on the operations preceding the exponential/sine layer is much stronger (polynomial) than the requirement on the subsequent operations (algebraic). As Theorem 8 shows, any non-polynomial operation preceding the sine operation and combined with linear operations restores the general finite approximability.

**General methods.**   All results presented in this paper that rule out global universality are essentially obtained by one of the three general methods: Khovansky's analytic theory of Pfaffian functions (Theorem 1), finding algebraic constraints (Theorems 3, 5, 6), and direct computations of limit functions (Theorems 4, 9, 10). It appears that all these methods are insufficient to conveniently separate the finite approximability class $\mathcal{G}_1 \setminus \mathcal{G}_2$ from the global approximability class $\mathcal{G}_2$. Only the last of the three methods seems useful for that, but it seems hard to find the full set of limit points for more complex families $H$ similarly to how we did that for the linear combinations of sines (Theorems 4) and the families $H^\sigma, H_N^{(5)}$ (Theorems 9, 10). It would be interesting to have a general method suitable for separating $\mathcal{G}_1 \setminus \mathcal{G}_2$ from $\mathcal{G}_2$.

**Limit points.**   Theorems 4 and 9 show that the nontrivial limit points of the families $H_N^{(2)}$ and $H^\sigma$ are some sort of resonances, reflecting some degeneracies of the model. The set of nontrivial limit points is small and explicit in these two cases. However, in a universal formula this set is the whole $C([0,1])$. It would be interesting to see how general is this dichotomy, e.g. whether the set of nontrivial limit points of some formula can be a small but not explicitly describable subset of $C([0,1])$.

**Neural networks.**   In Section 6 we described a general family of neural networks lying outside the class $\mathcal{G}_1$ and so non-universal. These networks can have a single layer of transcendental neurons (with activations such as $\sin, \tanh, e^{-x^2}$) as well as some layers with piecewise-polynomial activations. We have shown that though the resulting functions are only piecewise analytic, they still admit polynomial constraints. It remains, however, an open question whether networks with multiple transcendental layers involving the sine activation can be universal. Theorem 8 shows that such networks belong to $\mathcal{G}_1$, while Theorem 10 shows that the two-hidden-layer sine network of bounded complexity and with a single neuron in the second hidden layer belongs to $\mathcal{G}_1 \setminus \mathcal{G}_2$. We conjecture that, without $\arcsin$ or other special functions that help create periodic piecewise-linear dependencies, general $\sin$ networks of fixed complexity belong to $\mathcal{G}_1 \setminus \mathcal{G}_2$.

## Acknowledgments

The author thanks the anonymous reviewers for several useful suggestions. Research was supported by Russian Science Foundation, grant 21-11-00373.

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

# Contents

## A   Proof of Theorem 4

**Sufficiency (each function** (6) **is a limit point of** $H_N^{(2)}$**).**   First observe that each resonant component $x^m \sin(\omega x + h)$ is a uniform limit of $m + 1$ sine waves linearly combined to match the discretized derivative $\frac{d^m}{d\omega^m} \sin(\omega x + h - \pi m/2)$:

$$x^m \sin(\omega x + h) = \tfrac{d^s}{d\omega^m} \sin(\omega x + h - \pi m/2) \tag{13}$$

$$= \lim_{\Delta\omega \to 0} (\Delta\omega)^{-m} \sum_{n=0}^{m} (-1)^{m-n} \binom{m}{n} \sin((\omega + n\Delta\omega)x + h - \pi m/2). \tag{14}$$

Note also that any finite linear combination $\sum_{m=0}^{M-1} c_m \sin(\omega x + h_m)$ can be written as a single sine wave $c \sin(\omega x + h)$ with suitable amplitude $c$ and phase $h$. It follows that we can obtain any linear combination $\sum_{m=0}^{M-1} c_m x^m \sin(\omega x + h_m)$ as a uniform $\Delta\omega \to 0$ limit of linear combinations

$$\sum_{m=0}^{M-1} c_{m,\Delta\omega} \sin((\omega + m\Delta\omega)x + h_{m,\Delta\omega}) \tag{15}$$

with suitable values of $c_{m,\Delta\omega}$ and $h_{m,\Delta\omega}$. This shows in turn that the second component of expression (6),

$$\sum_{k=1}^{K}\sum_{m=0}^{M_k-1} c_{km}x^m \sin(\omega_k x + h_{km}),\tag{16}$$

can be obtained by limits of linear combinations of $\sum_{k=1}^{K} M_k$ sine waves with suitable phases and frequencies.

It remains to show that the first component of expression (6), $\sum_{m=0}^{2M_0-1} c_{0m}x^m$, can be obtained as a limit of linear combinations of $M_0$ sine waves. Observe that for any constant $x_0$

$$(x - x_0)^{2M_0-1} = \lim_{\Delta\omega\to 0} (\Delta\omega)^{2M_0-1} \sin^{2M_0-1}(\Delta\omega x - \Delta\omega x_0).\tag{17}$$

Since $2M_0 - 1$ is odd, $\sin^{2M_0-1}(\Delta\omega x - \Delta\omega x_0)$ is a linear combination of $M_0$ sine waves with frequencies $(2s - 1)\Delta\omega, s = 1, \ldots, M_0$ (and some phases). By a previously given argument, this implies that any linear combination of monomials $(x - x_0)^{2M_0-1}$ can also be obtained as a limit of some linear combinations of $M_0$ sine waves with frequencies $(2s - 1)\Delta\omega, s = 1, \ldots, M_0$. But the monomials $(x - x_0)^{2M_0-1}$ linearly span the space of polynomials of degree $\leq 2M_0 - 1$. It follows that any such polynomial can be obtained as a limit of linear combinations of $M_0$ sine waves, as claimed.

**Necessity (each limit point of $H_N^{(2)}$ has the from (6)).** We give two alternative proofs.

**Proof 1 ("clustering-based").** Suppose that a function $f_*$ is the uniform limit of functions $f_r \in H_N^{(2)}$. We identify the different *frequency clusters* associated with the sequence $f_r$. Let $0 \leq \omega_{1,r} \leq \ldots \leq \omega_{N,r}$ be the $N$ sorted frequencies appearing in the sine wave expansion of $f_r$; we assume here without loss of generality that all the frequencies are nonnegative. For each $n = 1, \ldots, N - 1$ one of the following three options holds:

1. either $\omega_{n+1,r} - \omega_{n,r} \overset{r\to\infty}{\longrightarrow} \infty$;
2. or $\sup_n(\omega_{n+1,r} - \omega_{n,r}) < \infty$;
3. or none of the above holds.

For each $n$, by taking a subsequence of $f_r$, we can exclude option 3. By taking subsequences $N$ times, we can then assume without loss of generality that for each $n$ either option 1 or option 2 holds. It follows that we can partition the set $\{1, \ldots, N\}$ into contiguous subsets $I_1 = \{1, 2, \ldots, i_1\}, I_2 = \{i_1 + 1, \ldots, i_2\}, \ldots, I_s = \{i_{s-1} + 1, \ldots, N\}$ such that for some constant $\Omega$

$$|\omega_{n,r} - \omega_{n',r}| \begin{cases} \overset{r\to\infty}{\longrightarrow} \infty, & \text{if } \nexists t : n, n' \in I_t; \\ \leq \Omega\ \forall r, & \text{if } \exists t : n, n' \in I_t. \end{cases}\tag{18}$$

The sets $I_t$ thus represent the largest clusters of frequencies that remain at a bounded distance from each other in the limit $r \to \infty$.

Clearly, there is at most one cluster whose frequencies remain bounded in the limit $r \to \infty$, namely $I_{t_0}$ with $t_0 = 1$. It will be convenient to assume that $I_{t_0}$ has bounded frequencies but possibly is empty.

For any $M = 1, 2, \ldots$ and $\Omega > 0$ let us denote by $H_{M,\Omega}^{(2)}$ the set of functions $f : [0, 1] \to \mathbb{C}$ of the form

$$f(x) = \sum_{n=1}^{M} c_n e^{i\omega_n x},\tag{19}$$

where $c_n \in \mathbb{C}$ are arbitrary amplitudes and $\omega_n \in [-\Omega, \Omega]$.

**Lemma 1.** *For any $f \in H_{M,\Omega}^{(2)}$ and $n = 1, 2, \ldots$*

$$\max_{x\in[0,1]} |f^{(n)}(x)| \leq 2^{M(M+1)}(1 + \Omega)^{M\max(n,M-1)} \max_{x\in[0,1]} |f(x)|.\tag{20}$$

We remark that the constant in Eq. (20) is likely suboptimal, but it must certainly grow with $M$, since, as the proof of the sufficiency part of Theorem 4 shows, arbitrarily narrow frequency bands are sufficient to generate any degree-$M$ polynomial.

*Proof.* The function $f$ satisfies the differential equation $Df = 0$, where

$$D = \prod_{n=1}^{M} (\tfrac{d}{dx} - i\omega_n) = \tfrac{d^M}{dx^M} - \sum_{n=0}^{M-1} b_n \tfrac{d^n}{dx^n} \tag{21}$$

with some constants $b_n$. Consider the vector-valued function $\mathbf{f}(x) = (f(x), f'(x), \dots, f^{(M-1)}(x))^T$. It satisfies the differential equation

$$\frac{d}{dx}\mathbf{f} = A\mathbf{f} \tag{22}$$

with the matrix $A = (a_{nm})_{n,m=0}^{M-1}$, where

$$a_{nm} = \begin{cases} 1, & n+1 = m \text{ and } n < M-1, \\ b_m, & n = M-1, \\ 0, & \text{otherwise.} \end{cases} \tag{23}$$

The solution of this equation is

$$\mathbf{f}(x) = e^{A(x-x_0)}\mathbf{f}(x_0). \tag{24}$$

If $\|\cdot\|$ is any norm on the $\mathbb{C}^M$, then

$$\|\mathbf{f}(x) - \mathbf{f}(x_0)\| \leq (e^{\|A\|\,|x-x_0|} - 1)\|\mathbf{f}(x_0)\|, \tag{25}$$

where $\|A\|$ is the respective matrix norm. Let $\|\cdot\|_\infty$ be the maximum ($l^\infty$) norm, then

$$\|A\|_\infty = \max_n \sum_m |a_{nm}| = \max\left(\sum_m |b_m|, 1\right) \leq \prod_{m=1}^{M}(1 + |\omega_m|) \leq (1+\Omega)^M. \tag{26}$$

Now let $x_0 = \arg\max_{x\in[0,1]} \|f(x)\|_\infty$ and let $n_0 = \arg\max_{n=0,\dots,M-1} |f^{(n)}(x_0)|$ so that $\|\mathbf{f}(x_0)\|_\infty = |f^{(n_0)}(x_0)|$. If $n_0 = 0$, then

$$\max_{x\in[0,1]} |f'(x)| \leq \max_{x\in[0,1]} |f(x)|. \tag{27}$$

Suppose that $n_0 > 0$. Observe that if $|x - x_0| \leq \tfrac{\ln 2}{2}(1+\Omega)^{-M}$, then

$$|f^{(n_0)}(x) - f^{n_0}(x_0)| \leq \|\mathbf{f}(x) - \mathbf{f}(x_0)\|_\infty \tag{28}$$

$$\leq (e^{\|A\|_\infty |x-x_0|} - 1)\|\mathbf{f}(x_0)\|_\infty \tag{29}$$

$$\leq (e^{(1+\Omega)^M |x-x_0|} - 1)|f^{(n_0)}(x_0)| \tag{30}$$

$$\leq \tfrac{1}{2}|f^{(n_0)}(x_0)|. \tag{31}$$

Consider the complex unit direction $z_0 = \frac{f^{(n_0)}(x_0)}{|f^{(n_0)}(x_0)|}$ and denote $l = \tfrac{\ln 2}{2}(1+\Omega)^{-M}$. For all $x$ such that $|x - x_0| \leq l$ we have

$$\Re(\bar{z}_0 f^{(n_0)}(x)) \geq \tfrac{1}{2}|f^{(n_0)}(x_0)|. \tag{32}$$

It follows that we can find in $[0,1]$ a subinterval of length $l/4$ on which either

$$\Re(\bar{z}_0 f^{(n_0-1)}(x)) \geq \tfrac{l}{2\cdot4}|f^{(n_0)}(x_0)| \tag{33}$$

or

$$\Re(\bar{z}_0 f^{(n_0-1)}(x)) \leq -\tfrac{l}{2\cdot4}|f^{(n_0)}(x_0)|. \tag{34}$$

Making another iteration, we can find in $[0,1]$ a subinterval of length $l/4^2$ on which either

$$\Re(\bar{z}_0 f^{(n_0-2)}(x)) \geq \tfrac{l}{4^2} \cdot \tfrac{l}{2\cdot4}|f^{(n_0)}(x_0)| \tag{35}$$

or

$$\Re(\bar{z}_0 f^{(n_0-2)}(x)) \leq -\tfrac{l}{4^2} \cdot \tfrac{l}{2\cdot4}|f^{(n_0)}(x_0)|. \tag{36}$$

Continuing in this way, for any $n = n_0, n_0 - 1, \ldots, 0$ we find in $[0, 1]$ a subinterval of length $l/4^{n_0-n}$ on which either

$$\Re(\bar{z}_0 f^{(n_0-n)}(x)) \geq \frac{l^{n_0-n}}{2^{(n_0-n)(n_0-n+1)+1}} |f^{(n_0)}(x_0)| \tag{37}$$

or

$$\Re(\bar{z}_0 f^{(n_0-n)}(x)) \leq -\frac{l^{n_0-n}}{2^{(n_0-n)(n_0-n+1)+1}} |f^{(n_0)}(x_0)|. \tag{38}$$

Taking $n = 0$, this implies

$$|f^{(n_0)}(x_0)| \leq 2^{n_0(n_0+1)+1} l^{-n_0} \max_{x \in [0,1]} |f(x)| \tag{39}$$

and so, by definition of $x_0$ and $n_0$, for any $n = 0, \ldots, M - 1$

$$\max_{x \in [0,1]} |f^{(n)}(x)| = |f^{(n_0)}(x_0)| \tag{40}$$

$$\leq 2^{n_0(n_0+1)+1} l^{-n_0} \max_{x \in [0,1]} |f(x)| \tag{41}$$

$$\leq 2^{(M-1)M+1}((2/\ln 2)(1+\Omega)^M)^{M-1} \max_{x \in [0,1]} |f(x)| \tag{42}$$

$$\leq 2^{M(M+1)}(1+\Omega)^{M(M-1)} \max_{x \in [0,1]} |f(x)|. \tag{43}$$

This proves the claim for $n \leq M - 1$. If $n \geq M$, then we use again Eq. (21) and the bound $\sum_m |b_m| \leq (1+\Omega)^M$. $\qquad\square$

**Lemma 2.** *The $L^\infty$ and $L^2$ norms are equivalent on the set $H_{M,\Omega}^{(2)}$: for any $f \in H_{M,\Omega}^{(2)}$*

$$\|f\|_2 \leq \|f\|_\infty \leq 2[2(1+\Omega)]^{M(M+1)/2}\|f\|_2 \tag{44}$$

*Proof.* Only the second inequality is nontrivial. By Lemma 1,

$$\|f'\|_\infty \leq c\|f\|_\infty, \quad c = [2(1+\Omega)]^{M(M+1)}. \tag{45}$$

It $x_0 = \arg\max_{x \in [0,1]} |f(x)|$, then for $|x - x_0| \leq 1/c$ we have $|f(x)| \geq (1 - |x - x_0|c)|f(x_0)|$ and so

$$\|f\|_2^2 \geq \int_0^{1/c} (1 - tc)^2 |f(x_0)|^2 dt = \tfrac{1}{3c}\|f\|_\infty^2, \tag{46}$$

implying the desired result. $\qquad\square$

Consider now the expansion of $f_r$ over different frequency clusters:

$$f_r = \sum_{t=1}^s f_{t,r}, \tag{47}$$

where $f_{t,r}$ denotes the $I_t$-component of $f_r$. We first show that we can discard the components associated with unbounded frequencies.

**Lemma 3.** *If $f_r$ converge to $f_*$ uniformly on $[0, 1]$, then $f_{t_0,r}$ also converge to $f_*$ uniformly on $[0, 1]$.*

*Proof. Step 1.* Let us first prove the statement in the case $f_* = 0$. Observe that each $f_{t,r}$ can be represented as

$$f_{t,r} = \sum_{q=\pm 1} e^{iq\omega_{t,r}x} g_{t,r,q}, \tag{48}$$

where $g_{t,r,1} = \overline{g_{t,r,-1}}$ and $g_{t,r,q} \in H_{2N,\Omega}^{(2)}$ with some $\Omega$ independent of $t$ and $r$. We use this representation for all $t$ except $t = t_0$ (since $f_{t_0,r} \in H_{2N,\Omega}^{(2)}$ already):

$$f_r = f_{t_0,r} + \sum_{t \neq t_0} \sum_{q=\pm 1} e^{iq\omega_{t,r}x} g_{t,r,q}. \tag{49}$$

Then,

$$\|f_r\|_2^2 = \langle f_r, f_r \rangle = \|f_{t_0,r}\|_2^2 + \sum_{t \neq t_0}^{s} \sum_{q=\pm 1} \|g_{t,r,q}\|_2^2 + \text{cross-terms}. \tag{50}$$

Observe that a product of $g, h \in H_{M,\Omega}^{(2)}$ belongs to $H_{M^2,2\Omega}^{(2)}$. It follows that each cross-term can be written as

$$\langle e^{i\Delta\omega_r}, g_r \rangle, \tag{51}$$

where $\Delta\omega_r \xrightarrow{r \to \infty} \infty$ and $g_r \in H_{(2N)^2,2\Omega}^{(2)}$; the function $g_r$ being a product of two functions out of the collection $F = \{f_{t_0,r}\} \cup \{(g_{t,r,q})_{t \neq t_0, q = \pm 1}\}$. We have

$$|\langle e^{i\Delta\omega_r}, g_r \rangle| = \left| \int_0^1 e^{-i\Delta\omega_r} g_r(x) dx \right| \tag{52}$$

$$= |\Delta\omega_r|^{-1} \left| \int_0^1 g_r(x) de^{-i\Delta\omega_r} \right| \tag{53}$$

$$\leq |\Delta\omega_r|^{-1} \left( 2\|g_r\|_\infty + \left| \int_0^1 e^{-i\Delta\omega_r} g_r'(x) dx \right| \right) \tag{54}$$

$$\leq |\Delta\omega_r|^{-1} (2 + c) \|g_r\|_\infty, \tag{55}$$

where we used the inequality $\|g_r'\|_\infty \leq c\|g_r\|_\infty$ with a constants $c = c(N, \Omega)$ as established in Lemma 1. Observe that if $g_r$ is a product of two functions from the collection $F$, say $g, h$, then we have

$$\|g_r\|_\infty \leq \|g\|_\infty \|h\|_\infty \leq C\|g\|_2 \|h\|_2, \tag{56}$$

with another suitable constant $C = C(N, \Omega)$, by Lemma 2. It follows that the cross-terms in Eq. (50) are dominated by the explicitly written terms:

$$|\text{cross-terms}| = o\left( \|f_{t_0,r}\|_2^2 + \sum_{t \neq t_0}^{s} \sum_{q=\pm 1} \|g_{t,r,q}\|_2^2 \right), \quad r \to \infty. \tag{57}$$

Therefore, if $\|f_r\|_\infty = o(1)$ as $r \to \infty$, then we also have

$$\|f_{t_0,r}\|_\infty = O(\|f_{t_0,r}\|_2) = O(\|f_r\|_2) = O(\|f_r\|_\infty) = o(1) \tag{58}$$

and similarly all

$$\|g_{t,r,q}\|_\infty = o(1). \tag{59}$$

*Step 2.* Now we prove the result for a general $f_*$. If $\|f_r - f_*\|_2 \to 0$, then the sequence $f_r$ is Cauchy:

$$\lim_{r_1 \to \infty} \lim_{r_2 \to \infty} \|f_{r_1} - f_{r_2}\|_2 = 0. \tag{60}$$

We write

$$f_{r_1} - f_{r_2} = (f_{t_0,r_1} - f_{t_0,r_2}) + \sum_{t \neq t_0}^{s} \sum_{q=\pm 1} e^{iq\omega_{t,r_1}x} g_{t,r_1,q} - \sum_{t \neq t_0}^{s} \sum_{q=\pm 1} e^{iq\omega_{t,r_2}x} g_{t,r_2,q}. \tag{61}$$

Assuming that $r_2 = r_2(r_1)$ and grows sufficiently fast, the frequencies $\omega_{t,r_1}, \omega_{t,r_2}$ and $0$ (corresponding to the first term) will form disjoint clusters. We can then use the argument from Step 1 and conclude that all $\|g_{t,r,q}\|_\infty = o(1)$, implying the desired result. $\square$

The last lemma shows that any uniform limit of functions $f_r \in H_N^{(2)}$ is also a uniform limit of band-limited functions $f_r \in H_{2M,\Omega}^{(2)}$ for some $\Omega < \infty$. Any function $f_r \in H_N^{(2)}$ is a solution of the differential equation

$$D_r f_r = 0, \tag{62}$$

where

$$D_r = \prod_{n=1}^{N} \left( \frac{d}{dx} - i\omega_{nr} \right) \left( \frac{d}{dx} + i\omega_{nr} \right). \tag{63}$$

If the frequencies $\omega_{nr}$ are bounded, by compactness we can find their limit points $\omega_n$. We argue now that the limiting function $f_*$ is a solution of the corresponding limiting equation $Df = 0$ with

$$D = \prod_{n=1}^{N} \left( \tfrac{d}{dx} - i\omega_n \right) \left( \tfrac{d}{dx} + i\omega_n \right). \tag{64}$$

Indeed, such solutions have the form

$$\mathbf{f}(x) = e^{A(x-x_0)} \mathbf{f}(x_0), \tag{65}$$

(cf. Eq. (24)), while the functions $f_r$ can be written as

$$\mathbf{f}_r(x) = e^{A_r(x-x_0)} \mathbf{f}_r(x_0), \tag{66}$$

with respective matrices $A_r$. As $r \to \infty$, the matrices $A_r$ converge (by taking a subsequence, if necessary) to $A$, and, by Lemma 1, $\mathbf{f}_r(x_0) \to \mathbf{f}(x_0)$. It follows that the function $f_*$ indeed is representable in the form (65), i.e. is a solution of equation $Df = 0$. This immediately implies the claim of the theorem (with isolated zero limiting frequencies $\omega_n = 0$).

**Proof 2 ("discretization-based").** Our second proof essentially relies on discretizations of the function $f_*$ and will consist of two steps. In the first step we introduce the discretizations, show that they satisfy discrete difference equations, and describe general solutions of these equations. In the second step we analyze how discretizations at different length scales agree with each other and show that the continuity of $f_*$ enforces representation (6) in the zero spacing limit.

**Step 1: Discretization.** Suppose that the function $f_*$ is the uniform limit of functions $f_r \in H_N^{(2)}, r = 1, 2, \ldots$ Fix some $\Delta x > 0$ and consider the sequences $\mathbf{u}^* = (u_s^*)_{s=0}^{\lfloor 1/\Delta x \rfloor}$ and $\mathbf{u}^{(r)} = (u_s^{(r)})_{s=0}^{\lfloor 1/\Delta x \rfloor}$ obtained by restricting these functions to the points $s\Delta x$:

$$u_s^* = f_*(s\Delta x), \quad u_s^{(r)} = f_r(s\Delta x), \quad s = 0, 1, \ldots, \lfloor 1/\Delta x \rfloor. \tag{67}$$

Since $f_r \in H_N^{(2)}$, each sequence $\mathbf{u}^{(r)}$ can be written as a linear combination

$$u_s^{(r)} = \sum_{n=1}^{N} b_{nr} e^{i\omega_{nr}s\Delta x} + \overline{b_{nr}} e^{-i\omega_{nr}s\Delta x}, \quad s = 0, 1, \ldots, \lfloor 1/\Delta x \rfloor \tag{68}$$

with some complex amplitudes $b_{nr} \in \mathbb{C}$ and frequencies $\omega_{nr} \in \mathbb{R}$ (the horizontal bar denotes complex conjugation).

We want to perform now the limit $r \to \infty$ and derive a similar representation for $\mathbf{u}^*$. To this end observe that, up to the boundary values, a sequence of the form $u_s = be^{i\omega s\Delta x}$ satisfies the finite difference equation

$$(T - e^{i\omega\Delta x})\mathbf{u} = 0, \tag{69}$$

where $T$ is the left shift:

$$(Tu)_s = u_{s+1}. \tag{70}$$

Since each sequence $\mathbf{u}^{(r)}$ has the form of linear combination (68) and the operators $T - e^{i\omega\Delta x}$ commute, it follows that $\mathbf{u}^{(r)}$ satisfies the finite difference equation

$$D^{(r)}\mathbf{u}^{(r)} \stackrel{\text{def}}{=} \prod_{n=1}^{N} [(T - e^{i\omega_{nr}\Delta x})(T - e^{-i\omega_{nr}\Delta x})]\mathbf{u}^{(r)} = 0. \tag{71}$$

This equation holds up to the $2N$ boundary components $u_s^{(r)}$ that are moved by the shifts $T$ contained in $D^{(r)}$ outside the domain $[0, 1]$ of the functions $f_r$.

We would like to perform the limit $r \to \infty$ in this difference equation, but we cannot directly perform it in the exponents $i\omega_{nr}\Delta x$, since the frequencies $\omega_{nr}$ may be unbounded and not have any limit points. However, the values $e^{i\omega_{nr}\Delta x}$ belong to the unit circle in $\mathbb{C}$, which is a compact set. Therefore, we can choose a subsequence $r_t$ such that $e^{i\omega_{nr_t}\Delta x} \to z_n^*$ as $t \to \infty$, with some $z_n^* \in \mathbb{C}, |z_n^*| = 1$.

Performing this limit in the difference equation, we then see that the limiting sequence $\mathbf{u}^*$ satisfies a difference equation

$$D\mathbf{u}^* \stackrel{\text{def}}{=} \prod_{n=1}^{N} [(T - z_n^*)(T - \overline{z_n^*})]\mathbf{u}^* = 0. \tag{72}$$

A general solution of this equation is again a linear combination of sine waves, but with possible resonances due to repeated values $z_n^*$. Suppose that the numbers $z_n^*$ contain $K$ distinct nonreal values $z_1, \ldots, z_K$ with multiplicities $M_k \geq 1$. Additionally, let $M_\pm \geq 0$ be the multiplicities of the real values $z_\pm = \pm 1$ among the numbers $z_n^*$. We can then write the general real solution of equation (72) as

$$u_s^* = \sum_{m=0}^{2M_- -1} b_{-,m} s^m (-1)^s + \sum_{m=0}^{2M_+ -1} b_{+,m} s^m + \sum_{k=1}^{K} \sum_{m=0}^{M_k -1} s^m (b_{km} z_k^s + \overline{b_{km}} \overline{z_k}^s), \tag{73}$$

with $M_- + M_+ + \sum_{k=1}^{K} M_k = N$. Equivalently, we can write

$$\mathbf{u}^* = \sum_{m=0}^{2M_- -1} b_{-,m} \mathbf{v}_{-,m} + \sum_{m=0}^{2M_+ -1} b_{+,m} \mathbf{v}_{+,m} + \sum_{k=1}^{K} \sum_{m=0}^{M_k -1} (b_{km} \mathbf{v}_{km} + \overline{b_{km}} \overline{\mathbf{v}_{km}}), \tag{74}$$

where we have introduced the vectors $\mathbf{v}_{\pm,m}, \mathbf{v}_{km}, \overline{\mathbf{v}_{km}} \in \mathbb{C}^{\lfloor 1/\Delta x \rfloor +1}$ with components

$$(v_{\pm,m})_s = s^m (\pm 1)^s, \quad (v_{km})_s = s^m z_k^s, \quad (\overline{v_{km}})_s = s^m \overline{z_k}^s. \tag{75}$$

The coefficients $b_{km}$ and $b_{\pm,m}$ in expansions (73), (74) are determined uniquely as long as $\Delta x$ is sufficiently small:

**Lemma 4.** *Suppose that* $\Delta x \leq 1/(2N - 1)$ *so that* $\dim \mathbf{u}^* = \lfloor 1/\Delta x \rfloor + 1 \geq 2N$. *Then the $2N$ vectors* $\mathbf{v}_{\pm,m}$, $\mathbf{v}_{km}$ *and* $\overline{\mathbf{v}_{km}}$ *are linearly independent and the coefficients* $b_{\pm,m}, b_{km}$ *are uniquely determined.*

*Proof.* For $1 \leq q \leq M_k$, consider the finite difference operators $D_{k,q}$ obtained by removing the factor $(T - z_k)^q$ from the operator $D$ that appears in Eq. (72):

$$D_{k,q} = \frac{D}{(T - z_k)^q}. \tag{76}$$

When applied to $\mathbf{u}^*$, this operator kills all the components of the expansions (73), (74) except those associated with $z_k$ and having degree $m \geq M_k - q$. In particular,

$$(D_{k,1} u^*)_s = a_k b_{k,M_k -1} z_k^s \tag{77}$$

with the explicit coefficient

$$a_k = (M_k - 1)! (z_k - \overline{z_k})^{M_k} \prod_{k' \in \{\pm\} \cup \{1, \ldots, K\} \setminus \{k\}} ((z_k - z_{k'})(z_k - \overline{z_{k'}}))^{M_{k'}} \neq 0. \tag{78}$$

We can then find the coefficient $b_{k,M_k -1}$ from Eq. (77) if there is at least one $s$ for which it holds (recall that it is not applicable to a few boundary values of $s$). The highest power of the shift $T$ appearing in $D_{k,1}$ is $T^{2N-1}$, so such $s$ exists as long as $\dim \mathbf{u}^* \geq 2N$.

Having found $b_{k,M_k -1}$, we can subtract the corresponding component $b_{k,M_k -1} \mathbf{v}_{k,M_k -1}$ from $\mathbf{u}^*$ and apply to the remainder the operator $D_{k,2}$, which will kill all terms except $b_{k,M_k -2} \mathbf{v}_{k,M_k -2}$. This allows to determine the next coefficient $b_{k,M_k -2}$. Continuing this process and extending it to other $z_k$ and $z_\pm$, we uniquely recover all the coefficients $b_{\pm,m}, b_{km}$. The linear independence of the vectors $\mathbf{v}_{\pm,m}$, $\mathbf{v}_{km}$ and $\overline{\mathbf{v}_{km}}$ follows from this uniqueness. $\qquad \square$

**Step 2: The limit $\Delta x \to 0$.** We consider now discretizations with variable spacing $\Delta x$. Our goal is to show that expansions (73) lead to the desired representation (6) in the limit $\Delta x \to 0$. In the notation, we will occasionally add $\Delta x$ to the subscripts to reflect the dependence of various quantities on the discretization spacing.

It is convenient to rewrite expansion (73) by rescaling the coefficients via $b_{km} = c_{km}(\Delta x)^m$ and introducing frequencies $\omega_k$ and $\omega_-$ such that $z_k = e^{i\omega_k \Delta x}$ and $-1 = e^{i\omega_- \Delta x}$, so as to make the expansion less $\Delta x$-dependent:

$$u_s^* = f_*(s\Delta x) \tag{79}$$

$$= \sum_{m=0}^{2M_- - 1} c_{-,m}(s\Delta x)^m e^{i\omega_- s\Delta x} + \sum_{m=0}^{2M_+ - 1} c_{+,m}(s\Delta x)^m \tag{80}$$

$$+ \sum_{k=1}^{K} \sum_{m=0}^{M_k - 1} (s\Delta x)^m (c_{km} e^{i\omega_k s\Delta x} + \overline{c_{km}} e^{-i\omega_k s\Delta x}). \tag{81}$$

The frequencies $\omega_k, \omega_-$ are determined from the values $z_{k,\Delta x}$ only up to integer multiples of $2\pi/\Delta x$:

$$\omega_k \in W_{k,\Delta w} \stackrel{\text{def}}{=} \left\{ \frac{-i\ln(z_{k,\Delta x}) + 2\pi l}{\Delta x} \right\}_{l\in\mathbb{Z}}, \tag{82}$$

$$\omega_- \in W_{-,\Delta w} \stackrel{\text{def}}{=} \left\{ \frac{\pi + 2\pi l}{\Delta x} \right\}_{l\in\mathbb{Z}}. \tag{83}$$

Observe that expansion (79)-(81) is invariant under downscaling, in the following sense. Since $s$ appears in it only through the combination $s\Delta x$, if this expansion holds for some $\Delta x$, then it also holds for any integer multiple $\Delta x' = l\Delta x$, with the same coefficients $c_{\pm,m}, c_{km}$ and frequencies $\omega_-, \omega_k$.

To discuss this effect in more detail, we argue now that we can establish a correspondence between different values $z_k$ across different scales. In general, downscaling is accompanied by aliasing: if the difference equation (72) holds with the values $z_{n,\Delta x}^*$ on the length scale $\Delta x$, then on the integer multiple length scale $l\Delta x$ it holds with the values

$$z_{n,l\Delta x}^* = (z_{n,\Delta x}^*)^l. \tag{84}$$

In particular, the values $z_{k,\Delta x}$ that are distinct on the length scale $\Delta x$ may collide when mapped to the length scale $l\Delta x$. However, in our setting we can bound the number of collisions thanks to the finiteness of the set of frequencies.

Specifically, consider length scales of the form $2^{-t}\Delta x$, with $t = 0, 1, \ldots$ The total number of collisions over all $t$ is bounded, since there are at most $K = N$ distinct values $z_k$. It follows that at sufficiently large $t$ there are no collisions, so that the number $K = K_t$ of distinct values $z_{k,2^{-t}\Delta x}$ is the same for all $t$, and we can enumerate these values so that for $t_2 > t_1$

$$z_{k,2^{-t_1}\Delta x} = z_{k,2^{-t_2}\Delta x}^{2^{t_2 - t_1}}. \tag{85}$$

Moreover, observe that for sufficiently large $t$ there will be no $\omega_-$-component (corresponding to $z_- = -1$) in the (79)-(81). Indeed, since the square roots of $-1$ are not real, the presence of the $\omega_-$-component for some $t$ would mean that $K_{t+1} > K_t$. But such an increase is possible only for a finite number of $t$'s.

Similarly, the $\omega_+$-component of the expansion (79)-(81) must have the form

$$f_{*,0}(s\Delta x) = \sum_{m=0}^{2M_+ - 1} c_{+,m}(s\Delta x)^m \tag{86}$$

with the same $c_{+,m}$ for all sufficiently large $t$, since any change of this component with $t$ is accompanied by an increase in $K_t$. The function $f_{*,0}$ uniquely extends to a continuous function on the whole interval $[0, 1]$,

$$f_{*,0}(x) = \sum_{m=0}^{2M_+ - 1} c_{+,m} x^m, \tag{87}$$

which agrees with the first component in the expansion (6).

This discussion shows that without loss of generality (by subtracting $f_{*,0}$ from $f_*$ if necessary), we may assume that expansion (79)-(81) contains only the last component, standing in line (81).

Assuming that transition from the length scale $\Delta x$ to $\Delta x' = l\Delta x$ occurs without collisions in $z_k$, the downscaling invariance condition can be written as

$$c_{km,\Delta x} = c_{km,\Delta x'}, \tag{88}$$

$$W_{k,\Delta x} \subset W_{k,\Delta x'}, \tag{89}$$

where the sets $W_{k,\Delta x'}$ are defined by Eq. (82). By Lemma 4, the coefficients $c_{km,\Delta x}$ are uniquely determined as soon as $\Delta x$ and $\Delta x'$ are small enough.

Considering again the length scales of the form $2^{-t}\Delta x$, we get a nested sequence

$$\ldots \subset W_{k,2^{-2}\Delta x} \subset W_{k,2^{-1}\Delta x} \subset W_{k,\Delta x}. \tag{90}$$

All the inclusions here are strict. The intersection

$$W_{k,\lim} = \cap_{t=0}^{\infty} W_{k,2^{-t}\Delta x} \tag{91}$$

is either empty or consists of a single frequency $\{\omega_k\}$. In the latter case we can again (as previously with $f_{*,0}$) identify the respective component of the function $f_*$:

$$f_{*,k}(x) = \sum_{m=0}^{M_k-1} x^m (c_{km} e^{i\omega_k x} + \overline{c_{km}} e^{-i\omega_k x}). \tag{92}$$

Our goal is to prove the that the intersections $W_{k,\lim}$ are indeed nonempty for all $k$. We will obtain this as a consequence of the continuity of the function $f_*$, which in turn follows from the fact that $f_*$ is assumed to be a uniform limit of continuous functions $f_r$.

To this end, observe that when upscaling from $2^{-t}\Delta x$ to $2^{-t-1}\Delta x$, the value $z_{k,2^{-t-1}\Delta x}$ is one of the two square roots of $z_{k,2^{-t}\Delta x}$. The case $W_{k,\lim} \neq \emptyset$ corresponds to the case when for all sufficiently large $t$ we take the square root with positive real part, so that

$$z_{k,2^{-t}\Delta x} \overset{t\to\infty}{\longrightarrow} 1. \tag{93}$$

In contrast, in the case $W_{k,\lim} = \emptyset$, for any $t_0$ there is $t > t_0$ such that $z_{k,2^{-t-1}\Delta x}$ has a negative real part, so that

$$z_{k,2^{-t}\Delta x} \overset{t\to\infty}{\nrightarrow} 1. \tag{94}$$

Now consider the vector $\mathbf{u}^*$ obtained by sampling the function $f_*$ at the points $s\Delta x$ as in Eq. (67). Consider the sequence $\mathbf{u}^{(*,t)}$ of vectors of the same dimension, obtained by sampling the function $f_*$ with additional shifts $2^{-t}\Delta x$:

$$u_s^{(*,t)} = f_*((s + 2^{-t})\Delta x), \quad s = 0, 1, \ldots, \dim \mathbf{u}^* - 1. \tag{95}$$

Since $f_*$ is continuous,

$$\mathbf{u}^{(*,t)} \overset{t\to\infty}{\longrightarrow} \mathbf{u}^*. \tag{96}$$

Recall that the vector $\mathbf{u}^*$ admits an expansion (74) over the vectors $\mathbf{v}_{\pm,m}, \mathbf{v}_{km}, \overline{\mathbf{v}_{km}}$. As mentioned earlier, we can assume without loss of generality that there are no $\mathbf{v}_{\pm,m}$ terms:

$$\mathbf{u}^* = \sum_{k=1}^{K} \sum_{m=0}^{M_k-1} (b_{km}\mathbf{v}_{km} + \overline{b_{km}}\,\overline{\mathbf{v}_{km}}). \tag{97}$$

Observe now that an analogous expansion can also be written for the shifted vectors:

$$\mathbf{u}^{(*,t)} = \sum_{k=1}^{K} \sum_{m=0}^{M_k-1} (b_{km}\mathbf{v}_{km}^{(t)} + \overline{b_{km}}\,\overline{\mathbf{v}_{km}^{(t)}}). \tag{98}$$

Moreover, the vectors $\mathbf{v}_{km}^{(t)}$ are either exact or approximate multiples of the vectors $\mathbf{v}_{km}$:

$$\mathbf{v}_{km}^{(t)} = \begin{cases} z_{k,2^{-t}\Delta x}\mathbf{v}_{km}, & m = 0, \\ z_{k,2^{-t}\Delta x}\mathbf{v}_{km} + o(1) & (t \to \infty), \quad m > 0. \end{cases} \tag{99}$$

By Lemma 4, the vectors $\mathbf{v}_{km}$ and $\overline{\mathbf{v}_{km}}$ are linearly independent for sufficiently small $\Delta x$, so convergence (96) requires all those coefficients $z_{k,2^{-t}\Delta x}$ for which $b_{km} \neq 0$ at least for one $m$ to converge to 1 (as in Eq. (93)). This proves that all $W_{k,\lim}$ are non-empty for such $k$. If for some $k$ we have $b_{km} = 0$ for all $m$, then this $k$-component is effectively missing and can be ignored. This completes the proof of the theorem.

# B   Proof of Theorem 5

Our proof will consist of three steps. In the first step we show that the polynomial constraint claimed in the theorem holds for functions of the form $e^{P(x)}$. In the second step we show that if polynomial constraints hold for the arguments of an algebraic function, then its values also admit a polynomial constraint. In the final third step we combine these results to complete the proof of the theorem.

We remark that our arguments are close to the arguments of the theory of algebraic-transcendent functions [16, 17], but we work in a discretized setting of uniformly sampled data without considering any derivatives.

**Step 1: Polynomial constraints for exponentials of polynomials.**   Consider the function

$$g(x) = e^{P(x)}, \tag{100}$$

where $P$ is a (possibly complex-valued) polynomial of degree $\leq d$. Consider operators $D_h^s$ of forward discrete derivatives of order $s$ with step $h$:

$$D_h^s g(x) = \tfrac{1}{h}(D_h^{s-1} g(x+h) - D_h^{s-1} g(x)) = \ldots = h^{-s} \sum_{k=0}^{s} (-1)^{s-k} \binom{s}{k} g(x+kh). \tag{101}$$

We have $D_h^{d+1} P = 0$ for any $h \in \mathbb{C}$ and any polynomial $P$ of degree $\leq d$ :

$$\sum_{k=0}^{d+1} (-1)^{d+1-k} \binom{d+1}{k} P(x+kh) = 0. \tag{102}$$

Separating the positive and negative coefficients, we can write this, equivalently, as

$$\sum_{k=0}^{\lfloor (d+1)/2 \rfloor} \binom{d+1}{2k} P(x+2kh) = \sum_{m=0}^{\lfloor d/2 \rfloor} \binom{d+1}{2m+1} P(x+(2m+1)h). \tag{103}$$

Exponentiating, it follows for $g(x)$ given by Eq. (100) that

$$\prod_{k=0}^{\lfloor (d+1)/2 \rfloor} (g(x+2kh))^{\binom{d+1}{2k}} - \prod_{m=0}^{\lfloor d/2 \rfloor} (g(x+(2m+1)h))^{\binom{d+1}{2m+1}} = 0. \tag{104}$$

We can view this as a polynomial constraint of the form

$$R(g(x), g(x+h), \ldots, g(x+(d+1)h)) = 0 \tag{105}$$

with some fixed polynomial $R$ of $d+2$ variables.

**Step 2: Extension of polynomial constraints to algebraic functions.**

**Lemma 5.**

1. *Suppose that $Q(y_0, y_1, \ldots, y_N)$ is an algebraic function, and $g_0(x), \ldots, g_N(x)$ are functions satisfying polynomial relations*

$$R_n(g_n(x), g_n(x+h), \ldots, g_n(x+sh)) = 0, \quad \forall x, h. \tag{106}$$

   *Let $g(x) = Q(g_0(x), \ldots, g_N(x))$ and $m \geq s(N+1)$. Then there exists a nontrivial polynomial $R$ of $m+1$ variables such that*

$$R(g(x), g(x+h), \ldots, g(x+mh)) = 0, \quad \forall x, h. \tag{107}$$

2. *Moreover, suppose that the function $Q$ is not fixed, and is only constrained by its degree $\deg Q \leq q$ and the degrees $\deg \phi_k \leq r$ of the polynomials appearing in its definition. Let $m \geq (q+1)\binom{N+r+1}{N+1} + s(N+1)$. Then there exists a nontrivial polynomial $R$ of $m+1$ variables such that relation (107) holds for all such $Q$.*

*Proof.* 1. Recall the defining relation (7) of the algebraic function $Q$ :

$$\sum_{k=0}^{q} \phi_k(z_0, \ldots, z_N) Q^k(z_0, \ldots, z_N) = 0. \tag{108}$$

We can use this relation to express $Q^q$ in terms of lower powers $Q^0, \ldots, Q^{q-1}$:

$$Q^q(z_0, \ldots, z_N) = -\phi_q^{-1}(z_0, \ldots, z_N) \sum_{k=0}^{q-1} \phi_k(z_0, \ldots, z_N) Q^k(z_0, \ldots, z_N). \tag{109}$$

By iterating this relation, for any power $t \geq q$ we can express $Q^t$ in terms of the powers $Q^0, \ldots, Q^{q-1}$:

$$Q^t(z_0, \ldots, z_N) = \phi_q^{-t}(z_0, \ldots, z_N) \sum_{k=0}^{q-1} \phi_{t,k}(z_0, \ldots, z_N) Q^k(z_0, \ldots, z_N), \tag{110}$$

where $\phi_{t,k}$ are suitable polynomials of degree not greater than $t \max_{k=0,\ldots,q} \deg \phi_k$. Substituting

$$z_n = g_n(x),$$

we get

$$Q^t(\mathbf{g}(x)) = \phi_q^{-t}(\mathbf{g}(x)) \sum_{k=0}^{q-1} \phi_{t,k}(\mathbf{g}(x)) Q^k(\mathbf{g}(x)), \tag{111}$$

where for brevity we have denoted $\mathbf{g}(x) = (g_0(x), \ldots, g_N(x))$.

Formula (111) remains valid if we replace $x$ by $x + lh$ with any $l$. Consider various monomials in the variables $Q(\mathbf{g}(x + lh))$ for $l = 0, \ldots, m$ :

$$F_{\mathbf{t}} = \prod_{l=0}^{m} Q^{t_l}(\mathbf{g}(x + lh)) \tag{112}$$

$$= \Big( \prod_{l=0}^{m} \phi_q^{-t_l}(\mathbf{g}(x + lh)) \Big) \sum_{k_0=0}^{q-1} \cdots \sum_{k_m=0}^{q-1} \widetilde{\Phi}_{\mathbf{t},\mathbf{k}}(\mathbf{g}(x), \ldots, \mathbf{g}(x + mh)) \mathbf{Q}^{\mathbf{k}}, \tag{113}$$

where $\mathbf{t} = (t_0, \ldots, t_m), \mathbf{k} = (k_0, \ldots, k_m)$, $\widetilde{\Phi}_{\mathbf{t},\mathbf{k}}$ are some polynomials of degree not greater than $(m + 1)(\max_l t_l) \max_k \deg \phi_k$, and

$$\mathbf{Q}^{\mathbf{k}} = \prod_{l=0}^{m} Q^{k_l}(\mathbf{g}(x + lh)). \tag{114}$$

Our goal is to prove that for sufficiently large $m$, the monomials $F_{\mathbf{t}}$ are linearly dependent (with some coefficients independent of $x$ and $h$), which is exactly equivalent to the existence of a polynomial $R$ satisfying relation (107). This is done by showing a linear dependence of the expressions (113) with different $\mathbf{t}$.

Specifically, observe that, regardless of $\mathbf{t}$, expressions (113) contain only finitely many (namely, $q^{m+1}$) different components $\mathbf{Q}^{\mathbf{k}}$. Since there are infinitely many monomials $F_{\mathbf{t}}$, it follows already from this observation that there exist nontrivial vanishing linear combinations of the monomials $F_{\mathbf{t}}$ if the coefficients in these linear combinations are allowed to belong to the field of all rational functions of $\mathbf{g}(x), \ldots, \mathbf{g}(x + mh)$. However, we need to show that nontrivial vanishing linear combinations can be found even with constant coefficients.

To this end, we will further reduce the combinatorial complexity of expressions (113) by removing high powers of the variables $g_n(x + lh)$ at the expense of introducing new rational functions of the variables $g_n(x + l'h)$ with $l' > l$. Recall that each $g_n$ is subject to polynomial relation (106). We can use these relations to transform polynomial functions of $\mathbf{g}(x)$ into rational functions of $\mathbf{g}(x), \mathbf{g}(x + h), \ldots, \mathbf{g}(x + sh)$ that contain only bounded powers of $\mathbf{g}(x)$. Specifically, isolate the variables $g_n(x)$ in the polynomials $R_n$ by writing equations (106) in the form

$$\sum_{j=0}^{d_n} \psi_{n,j}(g_n(x + h), \ldots, g_n(x + sh))) g_n^j(x) = 0 \tag{115}$$

with suitable polynomials $\psi_{n,j}$ of the remaining variables $g_n(x+h), \ldots, g_n(x+sh)$. The value $d_n$ is the degree of $g_n(x)$ in this representation. Then, similarly to how we did this for $Q$, we can express

$$g_n^{d_n}(x) = -\psi_{n,d_n}^{-1}(g_n(x+h), \ldots, g_n(x+sh)) \sum_{j=0}^{d_n-1} \psi_{n,j}(g_n(x+h), \ldots, g_n(x+sh)))g_n^j(x).$$

$$(116)$$

By iterating this relation, for any $t \geq d_n$ we get

$$g_n^t(x) = \psi_{n,d_n}^{-t}(g_n(x+h), \ldots, g_n(x+sh)) \sum_{j=0}^{d_n-1} \psi_{n,j,t}(g_n(x+h), \ldots, g_n(x+sh)))g_n^j(x),$$

$$(117)$$

with some polynomials $\psi_{n,j,t}$ of degree not greater than $t \max_j \deg \psi_{n,j}$. This relation remains valid if we replace $x$ by $x+lh$ with any $l$:

$$g_n^t(x+lh) = \psi_{n,d_n}^{-t}(g_n(x+(l+1)h), \ldots, g_n(x+(l+s)h))) \tag{118}$$

$$\times \sum_{j=0}^{d_n-1} \psi_{n,j,t}(g_n(x+(l+1)h), \ldots, g_n(x+(l+s)h)))g_n^j(x+lh). \tag{119}$$

We can now transform expression (113) by first replacing there all powers $t \geq d_n$ of $g_n(x)$ with powers $t \leq d_n$ and rational functions of $g_n(x+h), \ldots, g_n(x+sh)$, then replacing powers $t \geq d_n$ of $g_n(x+h)$ with powers $t \leq d_n$ and rational functions of $g_n(x+2h), \ldots, g_n(x+(s+1)h)$, and so on. We continue this process while the involved variables $g(x+lh)$ obey the condition $l \leq m$. As a result, we obtain a rational function of the variables $\mathbf{g}(x), \ldots, \mathbf{g}(x+mh)$ that contains only powers of the variables $g_n(x), \ldots, g_n(x+(m-s)h)$ not exceeding $d_n - 1$.

To establish the desired linear dependence, we want the rational functions appearing in these transformations of the monomials $F_{\mathbf{t}}$ to have the same denominator for all $\mathbf{t}$. It is convenient to ensure this in the following way. Choose some large $T$ and assume that all the powers $t_k$ in the monomials $F_{\mathbf{t}}$ vary between 0 and $T$, so that in total we have $(T+1)^{m+1}$ monomials. In expression (113), instead of the denominator $\prod_{l=0}^m \phi_q^t(\mathbf{g}(x+lh))$ use the common denominator

$$Z_0 = \prod_{l=0}^m \phi_q^T(\mathbf{g}(x+lh)). \tag{120}$$

This changes the polynomials $\widetilde{\Phi}_{\mathbf{t},\mathbf{k}}$ :

$$F_{\mathbf{t}} = Z_0^{-1} \sum_{\mathbf{k}} \Phi_{\mathbf{t},\mathbf{k}}(\mathbf{g}(x), \ldots, \mathbf{g}(x+mh))\mathbf{Q}^{\mathbf{k}}; \tag{121}$$

the new polynomials $\Phi_{\mathbf{t},\mathbf{k}}$ now have degrees not exceeding $(m+1)T \max_k \deg \phi_k$. Now we perform the iterative substitutions of the powers $g_n^t(x+km)$ in the polynomials $\Phi_{\mathbf{t},\mathbf{k}}$ as described above, while keeping the same denominator for all $\mathbf{t}$; this requires multiplying the common denominator by appropriate powers of $\psi_{n,d_n}(g_n(x+(l+1)h), \ldots, g_n(x+(l+s)h)))$. As a result of this process we obtain expressions

$$F_{\mathbf{t}} = Z^{-1} \sum_{\mathbf{k}} \mathbf{Q}^{\mathbf{k}} \sum_J \left( \prod_{\substack{n=0,\ldots,N \\ l=0,\ldots,m-s}} g_n^{J_{nl}}(x+lh) \right) \Psi_{\mathbf{t},\mathbf{k},J}(\mathbf{g}(x+(m-s+1)h), \ldots, \mathbf{g}(x+mh)).$$

$$(122)$$

Here $J = (J_{nl})$ is a matrix with elements $0 \leq J_{nl} \leq d_n - 1$, so summation over $J$ is finite. The polynomials $\Psi_{\mathbf{t},\mathbf{k},J}$ only depend on the remaining variables $\mathbf{g}(x+(m-s+1)h), \ldots, \mathbf{g}(x+mh)$.

Importantly, by taking into account previously observed bounds on the degrees of polynomials appearing in Eqs. (119) and (121), we observe that $\deg \Psi_{\mathbf{t},\mathbf{k},J} \leq CT$ with some constant $C$ depending on $s, m, N$ and the degrees of the polynomials $\phi_k$ and $\psi_{n,j}$ appearing in the assumed expansions (108) and (115).

Now the desired linear dependence of the monomials $F_{\mathbf{t}}$ results, with a suitable choice of $m$ and $T$, by comparing the number $(T + 1)^{m+1}$ of different monomials $F_{\mathbf{t}}$ with the number of different monomials that may appear in the expansions of the polynomials $\Psi_{\mathbf{t},\mathbf{k},J}$. Since these polynomials depend on the $s(N + 1)$ variables $\mathbf{g}(x + (m - s + 1)h), \ldots, \mathbf{g}(x + mh)$ and have degrees not exceeding $CT$, the number of different monomials in them does not exceed $C_1 T^{s(N+1)}$ with some constant $C_1$ depending on the same parameters as $C$. Taking into account that the summations over $\mathbf{k}$ and $J$ in expansion (122) involve, respectively, $q^{m+1}$ and $\prod_{n=0}^{N} d_n^{m-s+1}$ terms, we see that the $(T + 1)^{m+1}$ monomials $F_{\mathbf{t}}$ belong to a $C_1 q^{m+1}(\prod_{n=0}^{N} d_n^{m-s+1})T^{s(N+1)}$-dimensional linear functional space. Accordingly, if $s(N + 1) < m + 1$, then some of the monomials $F_{\mathbf{t}}$ will be linearly dependent if $T$ is large enough. This completes the proof of part 1 of the lemma.

2. The argument above produces nontrivial linear combinations of monomials $F_{\mathbf{t}}$ with coefficients generally dependent on $Q$. We show now that, assuming appropriate degree bounds for $Q$ and choosing a larger $m$, one can find nontrivial linear combinations with $Q$-independent coefficients. To this end, note that the main obstacle to choosing $Q$-independent coefficients is that the polynomials $\Psi_{\mathbf{t},\mathbf{k},J}$ in expansion (122) are $Q$-dependent. To overcome this obstacle, we express these polynomials both in the variables $\mathbf{g}(x + (m - s + 1)h), \ldots, \mathbf{g}(x + mh)$ and the coefficients $\mathbf{b}$ of the polynomials $\phi_k$:

$$\Psi_{\mathbf{t},\mathbf{k},J}(\mathbf{g}(x+(m-s+1)h),\ldots,\mathbf{g}(x+mh)) = \widetilde{\Psi}_{\mathbf{t},\mathbf{k},J}(\mathbf{b},\mathbf{g}(x+(m-s+1)h),\ldots,\mathbf{g}(x+mh)).$$
(123)

Here, the vector $\mathbf{b}$ includes all the coefficients of all $q + 1$ polynomials $\phi_k$ appearing in the defining identity (108). By retracing the steps to construct the polynomials $\Psi_{\mathbf{t},\mathbf{k},J}$, we see that their dependence on the coefficients $\mathbf{b}$ is indeed polynomial. Moreover, $\deg \widetilde{\Psi}_{\mathbf{t},\mathbf{k},J}$ is again bounded by $CT$, where $C$ is some constant possibly dependent on $N, s, m, q, \max \deg \phi_k, \max_j \deg \psi_{n,j}$, but not on $T$. Since we assume $\deg \phi_k \leq r$, we have $\dim \mathbf{b} \leq (q + 1)\binom{N+r+1}{N+1}$. The total number of variables in $\widetilde{\Psi}_{\mathbf{t},\mathbf{k},J}$ is thus not greater than

$$p = (q + 1)\binom{N + r + 1}{N + 1} + s(N + 1).$$
(124)

We repeat now the dimensionality-based linear dependence argument. The total dimensionality of polynomials $\widetilde{\Psi}_{\mathbf{t},\mathbf{k},J}$ considered at all $\mathbf{k}, J$ is $O(T^p)$. On the other hand, the number of monomials $F_{\mathbf{t}}$ is $(T + 1)^{m+1}$. If we set $m \geq p$, then for sufficiently large $T$ we can find a nontrivial vanishing linear combination of $F_{\mathbf{t}}$ with coefficients independent of $Q$.

$\square$

**Step 3: Proof of Theorem 5.** We apply now Lemma 5 to our setting in which $g_0(x) = x$ and $g_k(x) = e^{P_k(x)}, k = 1, \ldots, N$. We know from Step 1 that for $k = 1, \ldots, N$ conditions (106) hold with $s = \deg P_k + 1$. For $g_0$, condition (106) holds with $s = 2$ :

$$g_0(x) - 2g_0(x + h) + g_0(x + 2h) = 0.$$
(125)

Since we assumed that $\deg P_k \leq d$, we can set $s = \max(d, 1) + 1$ as a common value with which condition (106) holds for all $n = 0, \ldots, N$. The conclusion of Theorem 5 then follows from statement 2 of Lemma 5.

## C Proof of Theorem 6

Since the network has not more than $2^N$ hidden neurons and each of our activation functions has at most two branches, the output $g(x)$ of the network at any $x$ belongs to one of at most $2^N$ branches represented by analytic expressions. Consider each of these branches as defined for all $x \in [0, 1]$ and denote the family of all branches resulting from all $g \in H_N^{(4)}$ by $\widetilde{H}_N^{(4)}$. Observe that $\widetilde{H}_N^{(4)} \subset H_{N',q,r,d}^{(3)}$ if $N', q, r, d$ are large enough. Indeed, the input of any algebraic-exponential neuron is a polynomial of degree not greater than $d = 2^N$ (if the squared ReLU is present; otherwise, if only the step function and ReLU/leaky ReLU are present, then one can take $d = 1$). All the considered algebraic-exponential activations are rational functions of $x, e^x$ or $e^{\pm ix}$, and the subsequent neurons are polynomial, so any branch of the full network can be written in the form (8) with a rational $Q$ and $2N$ as the number of its exponential arguments. Rational $Q$'s have algebraic degree $q = 1$. The maximum degree of the

polynomials $\phi_k$ appearing in the defining relation (7) does not exceed $r = 2^N$ (again, if the squared ReLU is present; otherwise one can take $r = 1$). Summarizing,

$$\widetilde{H}_N^{(4)} \subset H_{2N,1,2^N,2^N}^{(3)}. \tag{126}$$

Theorem 5 implies now that there exists some $\widetilde{m}$ and a nonzero polynomial $\widetilde{R}$ of $\widetilde{m} + 1$ variables such that for any $\widetilde{g} \in \widetilde{H}_N^{(4)}$ and any size-$(\widetilde{m} + 1)$ arithmetic progression $\widetilde{a}, \widetilde{a} + \widetilde{h}, \ldots, a + \widetilde{m}\widetilde{h} \in [0, 1]$

$$\widetilde{R}(\widetilde{g}(\widetilde{a}), \widetilde{g}(\widetilde{a} + \widetilde{h}), \widetilde{g}(\widetilde{a} + 2\widetilde{h}), \ldots, \widetilde{g}(\widetilde{a} + \widetilde{m}\widetilde{h})) = 0. \tag{127}$$

We need to obtain a similar relation (11), but for $g \in H_N^{(4)}$. Each value $g(a + kh)$ in Eq. (11) belongs to one of the $2^N$ branches $\widetilde{g} \in \widetilde{H}_N^{(4)}$, which we associate with $p = 2^N$ colors. By applying van der Waerden's theorem with $p = 2^N$ and $s = \widetilde{m} + 1$, we see that if we set $m = N_{\text{vdW}}(\widetilde{m} + 1, 2^N) - 1$, then the length-$(m + 1)$ progression $a, a + h, \ldots, a + mh$ appearing in Eq. (11) contains at least one length-$(\widetilde{m} + 1)$ sub-progression $\widetilde{a}, \widetilde{a} + \widetilde{h}, \ldots, \widetilde{a} + \widetilde{m}\widetilde{h}$ on which the values of $g$ belong to the same branch. We can then satisfy condition (11) by defining the polynomial $R$ by

$$R(z_0, \ldots, z_m) = \prod_{\mathbf{s} = (s_0, \ldots, s_{\widetilde{m}})} \widetilde{R}(z_{s_0}, \ldots, z_{s_{\widetilde{m}}}), \tag{128}$$

where $\mathbf{s}$ runs over all length-$(\widetilde{m} + 1)$ progressions from the set $\{0, \ldots, m\}$.

## D  Proof of Theorem 8

The "only if" part follows from Theorem 5. The "if" part is a generalization of Lemma 1 from [29]. Suppose that $\sigma$ is not a polynomial; we want to prove that $H^\sigma \in \mathcal{G}_1$. By absorbing $h$ in $\sigma$, we can assume that $\sigma(0) \neq 0$. Dropping then $h$, we specialize to $f(x) = c \sin(\omega \sigma(bx))$. Consider an arbitrary finite subset $X = \{x_1, \ldots, x_N\} \subset [0, 1]$. By Theorem 2, it is sufficient to show that there exsts $b \in [-1, 1]$ such that the values $\sigma(bx_1), \ldots \sigma(bx_N)$ are rationally independent. Suppose that this is not the case. For fixed coefficients $\boldsymbol{\lambda} = (\lambda_1, \ldots, \lambda_N)$, the function $\sigma_{\boldsymbol{\lambda}}(b) = \sum_{n=1}^N \lambda_n \sigma(bx_n)$ is real analytic for $b \in [-1, 1]$. Since there are only countably many $\boldsymbol{\lambda} \in \mathbb{Q}^N$, there is some $\boldsymbol{\lambda}$ such that $\sigma_{\boldsymbol{\lambda}}$ vanishes at uncountably many $b \in [-1, 1]$. Then, by analyticity, $\sigma_{\boldsymbol{\lambda}} \equiv 0$ on $[-1, 1]$. Expanding this $\sigma_{\boldsymbol{\lambda}}$ into the Taylor series at $b = 0$, we get the identity $\sum_{n=1}^N \lambda_n x_n^m = 0$ for each $m$ such that $\frac{d^m \sigma}{dw^m}(0) \neq 0$. Suppose that $x_k$ have the natural order $0 \leq x_1 < \ldots < x_N$. If there are infinitely many $m$ for which $\frac{d^m \sigma}{dw^m}(0) \neq 0$, then, by letting $m \to \infty$, we first show that $\lambda_N = 0$, then $\lambda_{N-1} = 0$, etc. If $x_1 > 0$, then we exhaust all $\lambda_n$ in this way; otherwise we exhaust all except $\lambda_1$. However, $\lambda_1 = 0$ follows then from $\sigma(0) \neq 0$. Thus, either $\boldsymbol{\lambda} = 0$ or $\sigma$ is a polynomial, which contradicts our assumption.

## E  Proof of Theorem 9

Suppose that the function sequence $f_n(x) = c_n \sin(\omega_n \sigma(b_n x) + h_n)$ converges to a nontrivial limit point $f_*$. By a compactness argument, this requires at least one of the parameters $c_n, \omega_n$ to be unbounded. We expect unboundedness of one parameter to be compensated by another parameter converging to 0. Accordingly, we consider three possibilities: $\omega_n \to 0, \omega_n \to \omega \neq 0$ and $\omega_n \to \infty$. One of these possibilities can always by ensured by choosing a subsequence of $f_n$, if necessary (in the sequel we will without further mention repeatedly use such reductions based on compactness and/or transition to subsequences).

**Case 1:** $\omega_n \to 0$. Since $\sigma(b_n x)$ is uniformly bounded by our assumption and $\sin$ is periodic, we can assume that $\omega_n \sigma(b_n x) + h_n \overset{n \to \infty}{\longrightarrow} a$ with some constant $a \in [0, 2\pi)$, for all $x \in [0, 1]$. If $a \neq 0, \pi$, then the limit $f_*$ can only be trivial, a constant. Suppose that $a = 0$ (the case $a = \pi$ leads to analogous results). In this case $f_n(x) = c_n(\omega_n \sigma(b_n x) + h_n)(1 + o(1))$, so $f_*(x) = \lim_{n \to \infty} c_n(\omega_n \sigma(b_n x) + h_n)$. There are two possibilities: $b_n \to b \neq 0$ and $b_n \to 0$. In the first case we obtain nontrivial limits

$$f_*(x) = a\sigma(bx) + c, \quad a, b, c \in \mathbb{R}. \tag{129}$$

In the second case we can Taylor expand

$$\sigma(b_n x) = \sigma(0) + \tfrac{1}{r!} \tfrac{d^r \sigma}{dx^r}(0)(b_n x)^r + o(b_n^r), \quad \tfrac{d^r \sigma}{dw^r}(0) \neq 0. \tag{130}$$

A nontrivial $f_*$ requires $c_n b_n^r$ to have a finite nonzero limit. In this case, by adjusting $h_n$ we obtain nontrivial limits

$$f_*(x) = a_0 + a_r x^r, \quad a_0, a_r \in \mathbb{R}. \tag{131}$$

**Case 2:** $\omega_n \to \omega \neq 0$. Consider again the two possibilities: $b_n \to b \neq 0$ and $b_n \to 0$. In the first case we do not get any nontrivial limits. In the second case we again have Taylor expansion (130). Arguing as in Case 1, the only nontrivial limits that we can produce are of the form (131).

**Case 3:** $\omega_n \to \infty$. In this case we must have $b_n \to 0$, since otherwise $\sin(\omega_n \sigma(b_n x) + h_n)$ will have an unbounded number of alternating values $\pm 1$, so we must have $c_n \to 0$ and accordingly the only possible limit is $f_* = 0$. It follows that we again have expansion (130), and

$$f_n(x) = c_n \sin\left(\omega_n\big(\sigma(0) + \tfrac{1}{r!}\tfrac{d^r \sigma}{dx^r}(0)(b_n x)^r + o(b_n^r)\big) + h_n\right). \tag{132}$$

To avoid triviality due to an unbounded number of alternating values $\pm 1$ of sin, we must have $\omega_n b_n^r = O(1)$. We get new nontrivial limits of $f_n$ when $\omega_n b_n^r$ has a nonzero limit:

$$f_*(x) = c\sin(a_0 + a_r x^r), \quad a, a_0, a_r \in \mathbb{R}. \tag{133}$$

# F    Proof of Theorem 10

We need to prove that $H_N^{(5)} \in \mathcal{G}_1$ and $H_N^{(5)} \notin \mathcal{G}_2$.

**Proof of $H_N^{(5)} \in \mathcal{G}_1$.** For $N \geq 2$, this property can be concluded from Theorem 8, since we can just set $\sigma(x) = \sin(x)$ and $h = h\sin(0 \cdot x + \pi/2)$ in this theorem. In the case $N = 1$, observe that in the proof of Theorem 8 we only use $h$ to redefine $\sigma$ so that $\sigma(0) \neq 0$; otherwise we can set $h = 0$. Then, this proof becomes applicable to the family $f(x) = c\sin(\omega\sigma(bx))$ with $\sigma(x) = \sin(x + h)$ and any phase $h \neq \pi k, k \in \mathbb{Z}$; this is a subfamily of $H_{N=1}^{(5)}$.

**Proof of $H_N^{(5)} \notin \mathcal{G}_2$.** We will show that we cannot obtain a general continuous function as a limit of functions from $H_N^{(5)}$, because any such limit has a particular restrictive form.

We write $f_n(x) = c_n \sin(g_n(x)) + h_n$ and consider several cases depending on the behavior of $c_n$.

**Case 1:** $c_n \to 0$. In this case the limiting function is constant: $f_*(x) = h$.

**Case 2:** $c_n \to c \neq 0$. In this case $h_n$ must be bounded and hence can be assumed to converge to some $h$ (by possibly choosing a subsequence). Denote $\phi_n(y) = c_n \sin(y) + h_n$; then $\phi_n(y)$ converges uniformly on $\mathbb{R}$ to $\phi_*(y) = c\sin(y) + h$. Clearly, the functions $\widetilde{f}_n(x) = \phi_*(g_n)$ also uniformly converge to $f_*$.

Consider now two alternatives: either the family $\{g_n\}$ is uniformly equicontinuous, or not. In the first case, fix some point $x_0 \in [0,1]$, let $r_n = \lfloor \frac{g_n(x_0)}{2\pi} \rfloor$ and consider the shifted functions $\widetilde{g}_n(x) = g_n(x) - 2\pi r_n$. We have $\phi_*(g_n(x)) \equiv \phi_*(\widetilde{g}_n(x))$, the family $\{\widetilde{g}_n\}$ is also equicontinuous, and $\widetilde{g}_n(x_0) \in [0, 2\pi]$ for all $n$, so that the family $\{\widetilde{g}_n\}$ is additionally uniformly bounded on $[0,1]$. Then, by the Arzelà–Ascoli theorem, the family $\{\widetilde{g}_n\}$ contains a uniformly convergent subsequence. It follows that $f = \phi_*(g_*)$, where $g_*$ is a uniform limit of some functions from $H_{N+1}^{(2)}$. Theorem 4 describes all such limits.

In the case of the second alternative, there exist $\epsilon > 0$ and a subsequence $g_{n_s}$ such that for some $x_s, x_s' \in [0,1]$ we have $|x_s - x_s'| \to 0$ while $|g_{n_s}(x_s) - g_{n_s}(x_s')| \geq \epsilon$ for all $s$. Let $r_s = \lfloor \frac{g_{n_s}(x_s)}{2\pi} \rfloor$ and $\widetilde{g}_s(x) = g_{n_s}(x) - 2\pi r_s$. Then $\widetilde{g}_s(x_s) \in [0, 2\pi]$ and by choosing a subsequence we can assume that $g_s(x_s)$ converges to some $y_* \in [0, 2\pi]$. Using $|\widetilde{g}_s(x_s) - \widetilde{g}_s(x_s')| \geq \epsilon$ and the continuity of $\widetilde{g}_s$, we can then assume that $\widetilde{g}_s(x_s')$ converges to some $y_*' \neq y_*$. Also, we can assume that $x_s$ and $x_s'$ converge to

some $x_* \in [0,1]$. Then, by the uniform convergence of $\widetilde{f}_n$, $\sup_{x \in [x_s, x'_s]} |\widetilde{f}_{n_s}(x) - \widetilde{f}_*(x_*)| \to 0$ as $s \to \infty$. On the other hand, $\widetilde{f}_{n_s}(x) = \phi_*(g_{n_s}(x)) = \phi_*(\widetilde{g}_s(x))$ so that, by the continuity of $\widetilde{g}_s$,

$$\lim_{s \to \infty} \sup_{x \in [x_s, x'_s]} |\widetilde{f}_{n_s}(x) - \widetilde{f}_*(x_*)| \geq \sup_{y \in [y_*, y'_*]} |\phi_*(y) - \widetilde{f}_*(x_*)| \neq 0, \tag{134}$$

because $\phi_*$ in nonconstant on any interval $[y_*, y'_*]$. This contradiction shows that the second alternative is impossible.

**Case 3:** $c_n \to \infty$. In this case the range of $g_n$ must shrink, since otherwise the range of $f_n$ would be unbounded. Taking a subsequence, we can assume that for some $y_0 \in [0, 2\pi]$ we have $g_n(x) = y_0 + 2\pi m_n + o(1)$ uniformly on $[0,1]$. We can then Taylor expand

$$f_n(x) = h_n + c_n \sin(y_0) + c_n \cos(y_0)(g_n - y_0 - 2\pi m_n) \tag{135}$$

$$- \tfrac{c_n}{2} \sin(y_0)(g_n - y_0 - 2\pi m_n)^2 + c_n O((g_n - y_0 - 2\pi m_n)^3). \tag{136}$$

If $\cos(y_0) \neq 0$, then the respective term dominates the higher order terms and must be bounded as $n$ increases. We can then discard the higher order terms, and we see that $f_*$ is a uniform limit of the functions $c_n g_n - a_n$ with some $a_n$. These limits are among the limits of the family $H^{(2)}_{N+1}$, again covered by Theorem 4.

If $\cos(y_0) = 0$, then the respective term vanishes and the next, second order term will dominate the higher order terms. In this case $f_*$ is a limit of functions of the form $\psi_n = h_n + a_n(g_n - b_n)^2$. Substituting $g_n \in H^{(2)}_N$ and expanding products of sines, we see that $\psi_n \in H^{(2)}_{1+2(N+1)^2}$, so, again, the limits of $\psi_n$ are restricted by Theorem 4.

