# OpenReview forum: "Structure of universal formulas"
_NeurIPS.cc/2023/Conference — NeurIPS 2023 poster_

### Official Review · Reviewer_2cHj · 2023-06-25

**Soundness:** 3 good
**Presentation:** 3 good
**Contribution:** 2 fair
**Rating:** 6
**Confidence:** 4

**Summary:**

This paper studies function approximation in the spirit of the Kolmogorov-Arnold representation theorem.
A definition of expressivity classes is made which distinguishes pointwise and uniform approximation.
Function families with the form of neural networks are proposed and approximation results proven.

**Strengths:**

Definitions are systematic and arguments are clear.

**Weaknesses:**

As discussed at the bottom of p. 2, these constructions and arguments rely on infinite precision arithmetic
and are badly behaved when restricted to finite precision.  This is as expected for finite dimensional families
which live in the classes G_n of section 2.  The results are of mathematical interest but I do not see the interest
for the machine learning community.

**Questions:**

How could this line of work bear on or influence approximation theory which (like ref. [4]) is relevant for practical computation?

**Limitations:**

yes

---

> ### Author Rebuttal · Authors · 2023-08-05
>
> *How could this line of work bear on or influence approximation theory which (like ref. [4]) is relevant for practical computation?*
>
> Thank you for this reasonable question.
>
> 1. You mention Kolmogorov-Arnold, but the big difference between KA and our results is that the KA-type models include very complicated, non-elementary functions that do not naturally occur in real computations. In contrast, we consider only models that are explicitly defined by a finite number of elementary functions and arithmetic operations. Such models easily can/do occur in practice.
>
> 2. The effects we discuss are formulated irrespective of the limit of infinite magnitude/precision, and may potentially occur in realistic models. For example, we identify some functional families as belonging to the class $\mathcal {G}_0\setminus\mathcal {G}_1$ of constrained models that can shatter arbitrarily large finite sets, or to the class $\mathcal G_1\setminus\mathcal G_2$ of models that can approximate functions on any finite set but not on the whole domain. It is concievable that such deficiencies may occur in practice. You may call this conjecture a stretch, but practical interpretation of classical approximation theorems like Cybenko's [4] also involves various stretches. For example, Cybenko's theorem does not say anything about how many neurons, $10$ or $10^{10^{10}}$, are needed for a reasonable accuracy in a realistic problem. Rather, the value of this theorem is in showing that there are no fundamental constraints preventing single-hidden-layer networks from fitting generic functions; after that one just hopes that a moderate number of neurons is enough. Likewise, we see the main value of our results in showing that there are, or aren't, fundamental constraints present in a few natural functional families.
>
> 3. Moreover, our work can be viewed as clarifying conditions and conclusions of classical theorems like Cybenko's. This classical theorem guarantees approximability in the limit of infinitely many neurons, but does not rule out the possibility that the same approximability can be achieved with finitely many neurons. Imagine that such a strengthening of Cybenko's theorem would be true; wouldn't this have a profound effect on its interpretation in the context of ML? It turns out that this strengthening is not true in general for classical single-hidden-layer networks, but it does hold for relatively simple modified models. Our work sheds light on conditions necessary for such models.
>
> 4. Our class $\mathcal G_1\setminus\mathcal G_2$ is particularly appealing for practical interpretation. Its elements represent models that can fit any function on any finite set but cannot fit general continuous target functions on the full domain $[0,1]$. This implies that such models are trainable but poorly generalize: for a generic continuous target the deviation of the model from the target outside the training set does not decrease even if the size of the training set grows. In section 7 we give examples of sin-networks in this class, and our central conjecture is that general sin-networks with more than one hidden layer lie in this class. Now, our work was in fact partly motivated by an experimental observation that sin-networks tend to poorly generalize. Admittedly, we have not done any comprehensive experimental studies of this effect, but we typically observe that sin-network tend to overfit more than, say, ReLU networks whenever they train well on a training set (see examples in our [general author rebuttal](https://openreview.net/forum?id=gmVoaAxB1R&noteId=1hE4zF1UCW)), We think that our theoretical results contribute to a better understanding of this effect.

---

> > ### Comment · Reviewer_FCbX · 2023-08-11
> > **Rebtual 2 and 3 are inaccurate - Exact Rates are Known For Many Neural Network Classes**
> >
> > Though it is true that Hornik and Cybenko proved qualitative universal approximation theorems, ie density theorem; modern formulations are quantitative (so one knows explicit upper-bounds on the number of neurons required to approximate function in a given class).
> >
> > For example: [1] provide optimal rates for (uniformly) continuous function approximation on $[0,1]^d$ by ReLU feedforward neural networks, with optimal extensions to feedforward networks with general activation functions in [2].  For general quantitative statements, the key search word is "constructive approximation", e.g. see the book of DeVore and Lorentz on the topic.
> >
> > Concerning the pointwise topology, exact rates for real-valued ReLU neural network *interpolation* and not approximation, are known see [3], for multi-variate ReLU neural networks see Lemma 20 in [4], and for approximation in the pointwise topology, again for real-valued networks with sigmoidal activation function, see the main results of [5]; etc...
> >
> >
> >
> > [1] Shen, Zuowei, Haizhao Yang, and Shijun Zhang. "Optimal approximation rate of ReLU networks in terms of width and depth." Journal de
> > Mathématiques Pures et Appliquées 157 (2022): 101-135.
> >
> > [2] Zhang, Shijun, Jianfeng Lu, and Hongkai Zhao. "Deep Network Approximation: Beyond ReLU to Diverse Activation Functions." arXiv preprint arXiv:2307.06555 (2023).
> >
> > [3] Vardi, Gal, Gilad Yehudai, and Ohad Shamir. "On the optimal memorization power of relu neural networks." arXiv preprint arXiv:2110.03187 (2021).
> >
> > [4] Debarnot, Valentin, UNIBAS CH, and Ivan Dokmanic. "Small transformers compute universal metric embeddings." Journal of Machine Learning Research 24 (2023): 1-48.
> >
> > [5] Park, Sejun, et al. "Provable memorization via deep neural networks using sub-linear parameters." Conference on Learning Theory. PMLR, 2021.

---

> > > ### Author Response · Authors · 2023-08-13
> > > **Quantitative results and Unbounded precision**
> > >
> > >
> > > **Quantitative results.** To clarify: we never claimed that all network approximation theorems are qualitative. What we wrote is that Cybenko's theorem (and modern quantitative results we know of) do not allow to reasonably accurately estimate the network size required for specific real world problems. None of the papers cited by Reviewer FCbX theoretically predicts the network size necessary to, say, reach accuracy 90% on ImageNet [1,2]. A typical modern quantitative result says something like "if the target function belongs to a Hölder/Besov/Korobov etc. space, then a network of size/width/depth $N$ can generally approximate it with error $O(N^{-a})$", with some (possibly optimal) exponent $a$. There is a big gap between such theorems and real world applications, for multiple reasons. For example, the $O(..)$'s typically contain implicit large constants, making the bounds hardly useful for specific practical tasks. Four out of the five papers cited by Reviewer FCbX have these $O(..)$'s. Also, there is no simple way to assign a class like a specific Hölder space to a problem like ImageNet. There are, of course, other types of theoretical results (including some in the cited papers), but a significant application gap is present for all of them. This does not mean that these results are useless: their value is rather in revealing fundamental limitations and fundamental optimal designs of neural networks. We believe that our work shares this value.
> > >
> > > **Unbounded precision.** One of the papers cited by Reviewer FCbX - Park et al. (2021) - states that $O(N^{2/3})$ parameters are sufficient to memorize $N$ input-label pairs. Note that this result also, like our results, requires the parameters to have an unbounded precision: the $N$ input-label pairs carry at least $N$ bits of information, so the average amount of information per parameter is at least $N/O(N^{2/3})=\Omega(N^{1/3})\to \infty$. Moreover, the well-known standard and widely used bounds on the VC-dimension of neural networks with standard activations like ReLU [3,4] state that networks with $W$ parameters can shatter sets of size $\Omega(W^2)$ (see e.g. Theorem 3 in [4]). This again requires the precision of the weights to be unbounded since $W$ weights are used to store $\Omega(W^2)$ bits of information. So, unbounded parameter precision is a common and essential assumption in theoretical studies of neural networks.
> > >
> > >
> > > [1] https://www.image-net.org/
> > >
> > > [2] https://paperswithcode.com/sota/image-classification-on-imagenet
> > >
> > > [3] M. Anthony and P. Bartlett. Neural network learning: theoretical foundations. Cambridge University Press, 1999
> > >
> > > [4] Bartlett, P. L., Harvey, N., Liaw, C., & Mehrabian, A. (2019). Nearly-tight VC-dimension and pseudodimension bounds for piecewise linear neural networks. The Journal of Machine Learning Research, 20(1), 2285-2301.

---

> > > > ### Comment · Reviewer_FCbX · 2023-08-14
> > > >
> > > > This rebuttal alludes to their result clarifying gaps between the actual number of parameters in real world NN implementations.
> > > > However their paper does not as there is no connection to between these claims and papers results, since the paper does not provide any quantitative guarantees.

---

> > > > > ### Author Response · Authors · 2023-08-14
> > > > > **False statement**
> > > > >
> > > > > *This rebuttal alludes to their result clarifying gaps between the actual number of parameters in real world NN implementations.* This is an absolutely false statement by Reviewer FCbX, we nowhere claimed that.

---

> > > > > > ### Comment · Reviewer_FCbX · 2023-08-14
> > > > > >
> > > > > > I Quote:
> > > > > >
> > > > > > "You may call this conjecture a stretch, but practical interpretation of classical approximation theorems like Cybenko's [4] also involves various stretches. For example, Cybenko's theorem does not say anything about how many neurons, 10 or 10^10^10, are needed for a reasonable accuracy in a realistic problem."
> > > > > >
> > > > > > And
> > > > > >
> > > > > > "Moreover, our work can be viewed as clarifying conditions and conclusions of classical theorems like Cybenko's. This classical theorem guarantees approximability in the limit of infinitely many neurons, but does not rule out the possibility that the same approximability can be achieved with finitely many neurons. Imagine that such a strengthening of Cybenko's theorem would be true; wouldn't this have a profound effect on its interpretation in the context of ML?"
> > > > > >
> > > > > > Perhaps I am reading in to things to deeply, but these have undertones that somethibg quantitiave is being unveiled.  But not so much.  But perhaps I'm confused by suggestive writing.
> > > > > >
> > > > > >
> > > > > > ---
> > > > > >
> > > > > > In either case, the main issue is the claim about generalization, made both in the main rebutal and in response to one of my comments, which I prove to be false.  I think this is the main point the authors need to fix now.  Otherwise I have trouble seeing the utility of their results.

---

> > ### Comment · Reviewer_2cHj · 2023-08-16
> > **reply**
> >
> > Thanks to the authors for responding to my question.
> > Taking it into account along with the interest shown by the other reviewers, I increased my score.

---

### Official Review · Reviewer_Lm8e · 2023-07-07

**Soundness:** 4 excellent
**Presentation:** 4 excellent
**Contribution:** 3 good
**Rating:** 7
**Confidence:** 3

**Summary:**

The paper advances the basic understanding of the expressive power of fixed-complexity function families on [0, 1]. It defines three classes of function families: $\mathcal{G}_0$, whose members are infinite VC-dimension families, $\mathcal{G}_1$, whose members can approximately fit any finite number of points, and $\mathcal{G}_2$, whose members can achieve uniform approximation over the domain [0, 1]. Previous work has identified certain function families that belong to $\mathcal{G}_2$. This paper focuses on the difference sets, namely $\mathcal{G}_0 \setminus \mathcal{G}_1$ and $\mathcal{G}_1 \setminus \mathcal{G}_2$.

**Strengths:**

[originality]

- The paper explores the expressiveness of function classes that have fixed complexity and infinite VC dimension. To the best of my understanding, this research area remains largely unexplored beyond the prior results on universal formulas.

[quality]

- The mathematical framework is clearly defined, and the analysis is conducted rigorously.

[clarity]

- The paper is well-written.

[significance]

- The paper identifies certain specific function classes (such as $H^{(5)}$) that are seemingly complex and expressive but do not belong to $\mathcal{G}_2$ and belong to $\mathcal{G}_1$.

**Weaknesses:**

[originality]

- None in particular

[quality]

- None in particular

[clarity]

- None in particular

[significance]

- This paper only focuses on one-dimensional functions, which inevitably restricts the immediate applicability of the presented results to the field of machine learning. However, I consider this limitation to be a reflection of the fact that this area of analysis is still in its early stages.

**Questions:**

- Should Line 327 say $H_N^{(5)}$ instead of $H_N^{(2)}$?

[other minor suggestions]

- Line 353 “suggests” → “suggest”
- Line 183 “over the field Q” → “over the field Q of rational numbers” (as the field appears less frequently than R or Z in machine learning papers).
- Line 165 “is a subset” → “is a non-empty subset”

**Limitations:**

The paper acknowledges a limitation in its methodology, which is explained in the paragraph starting from Line 363. Specifically, the three methods used in the paper cannot effectively distinguish between $\mathcal{G}_1 \setminus \mathcal{G}_2$ and $\mathcal{G}_2$. As a result, the investigation in the paper is not completed and requires further research.

---

> ### Author Rebuttal · Authors · 2023-08-05
>
> Thank you for the careful reading and positive evaluation of our work! We agree with all your fixes, thanks!

---

> > ### Comment · Reviewer_Lm8e · 2023-08-11
> > **I have read the authors' response**
> >
> > Thank you for the response. I leave this comment to note that I have read it.

---

### Official Review · Reviewer_FCbX · 2023-07-18

**Soundness:** 3 good
**Presentation:** 2 fair
**Contribution:** 2 fair
**Rating:** 5
**Confidence:** 5

**Summary:**

The authors isolate the study universal hypothesis subclasses of $C([0,1])$ for the compact-open and point-open topologies respectively, and they compare universality here to infinite VC-dimension.  The results are interesting, though perhaps not that surprising nor novel, but are still publishable (in my opinion).

Some of the author's derivations could use more mathematical rigour and the terminology "formulas" is misleading as it alludes to logic/model theory, which does not appear in the paper.  The authors could also add some references to approximation theory in ML (some of which I provide below) to better tie in their results.  Details below in the Weakenesses section.

**Strengths:**

As stated above, the results are interesting, though perhaps not so surprising.

**Weaknesses:**

0. Add details to all proofs.  My main issue with this paper, is that many of its proofs contain handwavy and imprecise statements such as: 1) "Specifically, isolate the variables gn(x) in the polynomials" (line 739-740).  This is not rigorous and vague 2) equation (94) what space is this limit taken in; to me it seems to be $\mathbb{C}$ with its modulus as a norm, or 3) "by a classical theorem of Kronecker's" on lien 188; which one? (citation missing)... There are many such instances in derivation.

**Requested Change 0:**  Make all proofs rigorous and remove handwaved details (clear limits, clear steps, no missing references)

1. This sentence "especially dangerous from the generalization point of view" is purely speculative in the context of the paper, and one can doubt its accuracy [4] and [5] on benign overfitting for high-capacity models.

**Requested Change 1:** Since the authors only deal with approximation, they should remove this claim in the final version.

2. The hierarchy $G_1\subset G_2$ in (3) is misleading - The class $G_1$ is simply the set of subsets $A$ of $\mathbb{R}^{[0,1]}$ satisfying $A$ is dense in $\mathbb{R}^{[0,1]}$ for the point-open topology, which I denote $\tau_{PO}$; i.e. the topology of pointwise convergence (see [1] - Section 46, page 281).  So the inclusion of $G_1$ into $G_2$ is immediate and has a simple topological meaning; since $G_2$ is the same but for the compact-open topology (see [1] - Section 46, page 285).

From these remarks, and say Theorem 46.7 and Theorem 46.8 - both on page 281 of [1], imply $G_1\subset G_2$  and they give a topological interpretation of $G_1\subset G_2$.

I point this out because, from this perspective, one has a more general phenomenon.  Let us focus only on continuous functions.
Consider the standard poset $(\mathcal{T},\lesssim)$ of topologies on $C([0,1])$ with the partial order relation  $\tau_1\lesssim \tau_2$ if and only if $\tau_2$ is at-least as fine as $\tau_1$ (i.e.\ $\tau_1\subseteq \tau_2$).  For any $\tau\in \mathcal{T}$ define the class
$G_{\tau}:=\{A\subset C([0,1]): \bar{A}^{\tau} = C([0,1])\}$
where $\bar{A}^{\tau}$ denotes the closure of a subset $A$ of $C([0,1]$ with respect to the $\tau$ topology.  One immediately then deduces the contravariant-functorial relation: for all $\tau,\tilde{\tau}\in \mathcal{T}$
$\tau \lesssim \tilde{\tau} \Rightarrow G_{\tilde{\tau}}\subseteq G_{\tau}$.
So this type of construction holds in greater generality.  In fact, strictly finer topologies than the compact-open topology (i.e. the topology of uniform convergence since $[0,1]$ is compact and $\mathbb{R}$ is equipped with a metric - see Munkres Theorem 46.8) and it has been studied in the context of kernel methods [3] (see the strict topology) and non-uniform topologies on $L^1([0,1])$ were considered in [2] (see Theorems 1 and 2) in the context of deep learning.

**Requested Change 2:** Please add such comments circa (3), and emphasize that there are many such "classes" between $G_1$ and $G_2$, as well as, below or above then in the chain of inclusions (3).  Clearly the top of the chain consists of $G_{\tau_{disc}}$ where $\tau_{disc}$ is the discrete topology on $C([0,1])$ and the *correct* bottom of the chain consists of $G_{\{C([0,1]),\emptyset\}}$ (i.e. $\{C([0,1]),\emptyset\}$ is the trivial topology on $C([0,1])$.

3. Line 183 - You are considering $\mathbb{R}$ as an (infinite-dimensiona) $\mathbb{Q}$-module (vector space over the rationals).  This is not clear for readers who did not have a class or two on algebra during their studies.

**Requested Change 3:** Add an explanation of how $\mathbb{R}$ is an infinite-dimensional $\mathbb{Q}$-module.

4. Theorem 2 is not rigersouly stated.  What does "can approximate any R-valued function on X".  In reading the theorem's proof you must mean the point-open topology (which in this case consides with the compact-open topology, trivially).

**Requested Change 4:** Remove "can approximate any R-valued function on X" in the pointwise/point (in this case compact-open/uniform topologies.  Please write what is meant clearly, with "the usual for all $\epsilon$ theres exists an f satisfying .... " type statement.

Without this modification I cannot bump my score up to a pass, since the paper must be precise.

5. Missing Reference
**Requested Change 5:** Please add a precise reference to this hand-waving: "by classical Kronecker’s theorem" in the proof of Theorem 2 (it is classical if one has seen such results, but not in general).

6. Branching expressions terminology

**Requested Change 6:** Please call these piecewise functions.

---

I believe the manuscript will be in publishable condition when these changes are made.

**References**

[1] Munkres, James. "Topology James Munkres Second Edition."

[2] Kratsios, Anastasis, and Behnoosh Zamanlooy. "Do ReLU Networks Have An Edge When Approximating Compactly-Supported Functions?." Transactions on Machine Learning Research (2022).

[3] Chevyrev, Ilya, and Harald Oberhauser. "Signature moments to characterize laws of stochastic processes." The Journal of Machine Learning Research 23.1 (2022): 7928-7969.

[4] Bartlett, Peter L., et al. "Benign overfitting in linear regression." Proceedings of the National Academy of Sciences 117.48 (2020): 30063-30070.

[5] Tsigler, Alexander, and Peter L. Bartlett. "Benign overfitting in ridge regression." J. Mach. Learn. Res. 24 (2023): 123-1.


**Note** I did not use mathcal since open-review had compilation issues with it...

**Questions:**

- Related to requested change 2: is there a topology $\tau$ on $C([0,1])$ such that $\mathcal{G}_{\tau}=\mathcal{G}_0$?
- In the proof of Theorem 2, does one really need to index over $\mathbb{R}$; isn't $\mathbb{R}\setminus \mathbb{Q}$ enough?
- Why do you call these formulas?  Is there any connection to logic/model theory?  To me the correct/standard term is universal hypothesis/function classes, in the context of approximation theory and universal approximation theorems.

**Limitations:**

Not really, but this section doesn't really affect this paper.

---

> ### Author Rebuttal · Authors · 2023-08-06
>
> Thank you for the comprehensive reading of our paper including the proofs, and a positive evaluation of our work. We also appreciate your many comments, though we respectfully disagree with some of them.
>
> **Requested changes.**
>
> 0. We don't actually see what is imprecise in the examples you give, except for the reference for Kronecker's theorem which is indeed missing.
>
>     1) We don't see what is not rigorous and vague here. You cite the beginning of a sentence; the end of the same sentence specifies the exact sense in which the variable is isolated - namely, by expanding the polynomial as a univariate polynomial in this variable with coefficients in the ring of polynomials in the remaining variables (Eq. (115)).
>
>     2) Equation (94) is a statement on convergence of a sequence of complex numbers. There is a unique standard meaning of this convergence known from the basic calculus course; we don't see what is ambiguous here.
>
>     3) Indeed, a reference to Kronecker's theorem is missing. We will add it to the paper (the original Kronecker's paper is [1], and a recent survey can be found in [2]). Thank you for this suggestion.
>
> 1. We don't see why this sentence is speculative. By definition, the models in the class $\mathcal G_1\setminus\mathcal G_2$ can approximate any target function on any finite set, but, in general, cannot approximate a continuous target function on the whole domain $[0,1]$. This means that a model from this class can be fitted on any finite training set, but, in general, outside the training set the deviation from the target will not decrease to 0 as the size of the training set grows. This implies that the model can be fitted, but cannot properly generalize.
> We don't see why the papers on benign overfitting that you cite are relevant here - these papers deal with a completely different setting of high-dimensional linear models under special assumptions.
>
> 2. Thank you for this comment. We only consider the classes $\mathcal G_0, \mathcal G_1, \mathcal G_2$ in our paper because they have a simple meaning and because we prove something nontrivial about them. We don't claim that this hierarchy is exhaustive; of course, there are many other classes, in particular those induced by different topologies as you describe. However, we don't see a point in discussing them in the paper since we don't prove anything about them. Discussing topological and other issues would only distract the reader from the essense of our work and make our paper less accessible. As you can see from the neighboring review of Reviewer 2cHj, our work is already criticized for being too focused on abstract math.
>
> 3. Respectfully, adding terminology like "$\mathbb Q$-module" would not add any substance to our paper, but would certainly make it less accessible.
>
> 4. We don't see why Theorem 2 is not rigorously stated. After restriction to the finite set $X$, approximation becomes finite-dimensional and has a unique standard meaning.
>
> 5. Done, thank you.
>
> 6. Thank you for this suggestion, but we want to keep the term "branching expressions" because it reflects the actual branching involved in the computation of these expressions.
>
>
> We call our functional families "formulas" because they are defined by finite expressions composed of numbers, arithmetic operations and elementary functions. One can think of these formulas as words in a finite alphabet; this connects them to formulas in logic/model theory. It is important in our context that a model consists of a finite number of explicit standard operations. In contrast, general hypothesis/function classes are not that restrictive; even the class of single-hidden-layer neural networks $f(\mathbf x)=\sum_{n=1}^N c_n\sigma(\mathbf a_n\cdot \mathbf x+h_n)$ contains an operationally indefinite activation $\sigma$, not to mention classes like Sobolev spaces, etc.
>
>
> [1] L. Kronecker, Näherungsweise ganzzahlige Auflösung linearer Gleichungen, Monats. Königl. Preuss. Akad. Wiss. Berlin, 1179–1193 (1884), pp. 1271-1299
>
> [2] S. M. Gonek, H. L. Montgomery, Kronecker’s approximation theorem, Indagationes Mathematicae, Volume 27, Issue 2, 2016, Pages 506-523, (https://www.sciencedirect.com/science/article/pii/S0019357716000148)
>
> **Our question.** You write: *The results are interesting, though perhaps not that surprising nor novel*. If our results are not novel, can you please cite specific publications containing our theorems 4, 5, 6, 10 or their close analogs?

---

> > ### Comment · Reviewer_FCbX · 2023-08-11
> > **Response**
> >
> > **Retort**
> >
> > 1. Generalization is a probabilistic notion, which holds in expectation.  Let's consider a simple case where there is a "true" target function $f:X\mapsto \mathbb{R}$ (to be learnt), a probability measure $\mu$ on $X$ from which we draw samples, and suppose that the data-generating probability measure $\mathbb{P}$ on $X\times \mathbb{R}$ is noisless; meaning that $\mathbb{P}$ is the pushforward of $\mu$ by the map from $X$ to $X\times \mathbb{R}$ given for every $x\in X$ by $x\mapsto (x,f(x))$ .
> > In this case, the true risk of any $g:X\mapsto \mathbb{R}$ is
> >
> > $R(g) := \mathbb{E}^{(X,Y)\sim \mathbb{P}} [|g(X)-Y|]  = \mathbb{E}^{X\sim \mu}[|g(X)-f(X)|].$
> >
> >
> > Suppose that $f^{(0)},f^{(1)},\dots$ is a sequence of functions from $X$ to $\mathbb{R}$, e.g. in $G_1\setminus G_2$, which converge (only) pointwise to a limit $f$, also mapping $X$ to $\mathbb{R}$.
> > If suppose that there is some function $F\in L^1_{\mu}(X)$ satisfying such that, for every $n\in\mathbb{N}$, $\mathbb{E}^{X\sim \mu}[|f^{(n)}(X)|]$ and $\mathbb{E}^{X\sim \mu}[f(X)|]$ are all bounded-above by $\mathbb{E}^{X\sim \mu}[|F(X)|]$.
> >
> > Then the dominated convergence theorem the sequence $f_0,f_1,\dots$ converges to $f$ in $L^1_{\mu}(X)$.  This shows that a sequence of functions in $G_1\setminus G_2$, which only approximates the true/target function pointwise can make the true risk vanish; i.e. we have shown that $\operatorname{lim}_{n\mapsto \infty} R(f^{(n)})=0$ even if the sequence $f^{(0)},f^{(1)},\dots$ need not approximate $f$ uniformly.
> >
> >
> > 2. I agree with your rebuttal, that is a fair point.
> >
> > 3. Perhaps $\mathbb{Q}$-module could sound off-putting, but a $\mathbb{Q}$-vector space is exactly the same thing and all readers know what that is (since the all modules over a division ring are free modules; aka vector spaces).  So I respectfully disagree, I think this clarification should be included, but perhaps with the $\mathbb{Q}$-vector space terminology.
> >
> > 4. Here are two meanings.  For example, let $X$ be a finite set with at-least two elements.  Consider the (trivial) topology $\tau_1:=\{\emptyset, \mathbb{R}^X\}$ (note brackets did not compile), the topology of pointwise convergence $\tau_2$, ie the product topology on $\mathbb{R}^X$, and the (discrete) topology $\tau_3:=\mathbb{R}^X$, on $X$.  Approximation means density in $\mathbb{R}^X$ with respect to some topology, the authors mean $\tau_2$, but since they do not make any explicit statements one can't help onder if its $\tau_1$, $\tau_3$, or something even more exotic.
> > In short, I have provided at-least 3 non-unique meanings.  Please formulate it precisely.
> >
> > 5.  Thank you for adding the reference.
> >
> > 6. That makes sense.
> >
> >
> >
> > All other other points are minor.
> >
> >
> >
> > ---
> > **Comment about the terminology formulas:**
> >
> > With respect to the terminology "formulas", I see what angle you are getting at, but as far as I know, model theoretic descriptions of functions representable in a model/language are usually binary.  Namely, a function can be exactly expressed in a language with finitely many operations.  For example, O-minimal functions; see e.g. Wilkie's theorem.  Rather, the authors consider approximation problems, which seem unorthodox for model theory (but this last comment, I am not an expert on, so take it as a point of curiosity).
> >
> >
> > ---
> >
> > **Concerning the Typesetting of My Retort:**
> >
> > P.s.: Some notation, such as sequences and expectation look odd since I'm having trouble compiling LaTeX with subscripts on open review; apologies for the type setting.

---

> > > ### Author Response · Authors · 2023-08-15
> > >
> > > 1\. "Model generalization" is a general concept reflecting our expectation that if a model is fitted to a target function on a subset of the input domain, then the model should agree with the target on the whole domain. Specific meaning of "fitted", "agree", etc. is a matter of convention. In our remark we point out a particular sense (different from yours) in which the class $\mathcal G_1\setminus\mathcal G_2$ conflicts with generalizability. Our remark is not a theorem-like assertion; it is our interpretation of a mathematical object. The reader may like or dislike this interpretation, but we don't claim anything going beyond the basic definition of the class $\mathcal G_1\setminus\mathcal G_2$. The remark is useful for the paper, and we will not remove it. However, we agree to add the clause "in the uniform norm" in the end of the first sentence to avoid confusion with other kinds of approximation.
> > >
> > > 3\. We don't see why any clarification is needed here. Theorem 2 is a very simple and known fact that we already explain in detail for the reader's convenience. For comparison, in the machine learning paper [S] the same fact is described in three lines, even without mentioning Kronecker's theorem. We already agreed to supplement the reference to Kronecker's theorem by a specific citation as you asked.
> > >
> > > 4\. First, your topology $\tau_1$ is not a topology on $\mathbb R^X$, so it is unclear what it has to do with convergence of functions. Second, we don't introduce any topologies in our paper (we don't even use the word "topology"). Why would the reader expect that we use some exotic topology when there is a unique standard notion of approximation in $\mathbb R^X$ for a finite $X$? We don't see a point in making the paper more complicated than it needs to be.
> > >
> > > [S] Sontag, E. D. (1997). Shattering all sets of k points in “general position” requires (k—1)/2 parameters. Neural Computation, 9(2), 337-348.

---

> > > > ### Comment · Reviewer_FCbX · 2023-08-15
> > > >
> > > > 1. The definition of the term generalization is standard.  If you would like to adopt a non-standard meaning, then please use another word (we simply cannot redefined what a Banach space is at a whim now can we?).
> > > >
> > > > 3. Just because there exists a well-cited paper which is imprecise, it doesn't mean that we have to be here.
> > > >
> > > > 4. The curly brackets did not compile and indeed there was a typo, it should read: $\tau_1:=\{\emptyset,\mathbb{R}^X\}$. I aupologize for the typo, but with this correction (which I retroactively fixed), $\tau_1$ is indeed a topology on $\mathbb{R}^X$ (the so-called trivial or cofree topology) so has so much to do with convergence of functions.
> > > >
> > > > More importantly, my point here, is the following: you claimed there is a "unique canonical meaning" to your statement and I simply show there is several canonical meanings; and so not unique.  Thus, a clear rigerous satement is needed in the discussed theorem.

---

> > > > > ### Author Response · Authors · 2023-08-15
> > > > >
> > > > > "Generalization" is a general concept while "Banach space" is a precise mathematical term; these are completely different things.
> > > > >
> > > > > Here's the Wikipedia page on Banach space: https://en.wikipedia.org/wiki/Banach_space
> > > > > It includes (of course) a precise mathematical definition.
> > > > >
> > > > > Here's the Wikipedia pages on generalization in learning and machine learning:
> > > > >
> > > > > - https://en.wikipedia.org/wiki/Generalization_(learning)
> > > > > - https://en.wikipedia.org/wiki/Machine_learning#Generalization
> > > > >
> > > > > They don't include any formulas and in the specific machine learning context they give a conceptual description similar to ours: "*the goal of generalization: while optimization algorithms can minimize the loss on a training set, machine learning is concerned with minimizing the loss on unseen samples*".
> > > > >
> > > > > This example shows that the word "generalization" can be used in our general sense. You may be used to a particular definition of generalization, but this does not mean that we are not allowed to use this word in another sense.
> > > > >
> > > > > We don't agree with your other points, either. We believe that they all concern aspects of the paper that should be at the discretion of the authors.
> > > > >
> > > > > However, we agree to make changes that you request if any other reviewer confirms your points.
> > > > >
> > > > > **Our question.** Please answer our earlier question on citations confirming lack of novelty of our work. You write that our results are "*not that surprising nor novel*". Our main results are our theorems 4, 5, 6, 10. Which specific publications contain these or closely similar theorems?

---

> > > > > > ### Comment · Reviewer_FCbX · 2023-08-15
> > > > > > **Concluding Remarks on Discussion**
> > > > > >
> > > > > > **Points**
> > > > > >
> > > > > > I agree that the statements of those results are interesting.  My main issue is the following points:
> > > > > >
> > > > > >  - Imprecise statement of Theorem 2.
> > > > > >
> > > > > >  - Imprecise mathematical formulation of the Pfaffian class (as compared, e.g. Definition 2.1 of [1], which is clear and leaves no room for guessing).
> > > > > >
> > > > > >  - Claims about non-standard notions of generalization without comparing them to standard notions of risk, or PAC learnability from statistical learning theory.  Which, moreover, are provably false when "generalization" is understood to mean the standard statistical notion o risk (see above counter-example).
> > > > > >
> > > > > >
> > > > > > **Changes Required for Increase of Score**
> > > > > >
> > > > > > Though I feel that the results which the authors showcase are interesting, without addressing the above three points, I do not feel the paper is at the level of NeurIPS.  With those changes, I believe it is.
> > > > > >
> > > > > >
> > > > > > I feel the discussion has communicated my concerns to the authors, and all important information has now been interchanged between both parties. I look forward to, and do hope to see, those changes I requested implemented.
> > > > > >
> > > > > >
> > > > > > Thank you for the discussion,
> > > > > >
> > > > > > Reviewer FCbX
> > > > > >
> > > > > >
> > > > > > [1] Gabrielov, Andrei, and Nicolai Vorobjov. "Complexity of computations with Pfaffian and Noetherian functions." Normal forms, bifurcations and finiteness problems in differential equations 137 (2004): 211-250.

---

> > > > > > > ### Author Response · Authors · 2023-08-15
> > > > > > > **Our commitment**
> > > > > > >
> > > > > > > Thank you for the constructive position.
> > > > > > >
> > > > > > > **Our commitment.** We hereby confirm that we are not convinced by Reviewer FCbX in the necessity and usefulness of the three revisions requested by this reviewer, but we will be convinced and will happily make any of these three revisions in case they are also supported by any of the other reviewers.
> > > > > > >
> > > > > > > We believe that this our commitment resolves the issue in the most fair and positive way.
> > > > > > >
> > > > > > > Thanks again,
> > > > > > >
> > > > > > > The authors

---

### Official Review · Reviewer_BXX4 · 2023-07-28

**Soundness:** 4 excellent
**Presentation:** 4 excellent
**Contribution:** 3 good
**Rating:** 6
**Confidence:** 3

**Summary:**

The authors study "universal formulas" -- specifically, families of parametric
mappings $(f_{\omega})\_{\omega \in \Omega} \subset C([0, 1])$ where $\Omega$ is
a domain in a finite-dimensional euclidean space, such that $(f_{\omega})$ is
actually dense in $C([0, 1])$ with the sup-norm -- motivated in part by
applications to neural networks. In this connection, universal formulas should
be contrasted with typical "universal approximation" results for neural
networks that state that various fixed-depth neural networks can approximate
any continuous function on $[0,1]$ by *increasing the number of neurons*
sufficiently; universal formulas achieve this approximation with a fixed number
of neurons by instead exploiting (at least indirectly) unbounded-magnitude
parameters and infinite-precision arithmetic. The authors present a hierarchy
of nested expressiveness classes -- in decreasing order, families with infinite
VC dimension, families that achieve arbitrary approximation on arbitrary finite
sets, and families that achieve arbitrary approximation of arbitrary continuous
functions (i.e., universal formulas) -- and study obstructions to a family
being in a class higher in the hierarchy, but not a class lower in the
hierarchy. Starting from the classical example of the infinite VC-dim class
$\\{c \sin \omega x\\}_{c,\omega \in \mathbb{R}}$, the authors prove using
algebraic methods that a fairly general class of
"polynomially-exponentially-algebraic" expressions, which includes neural
networks with topology given by a (fixed size) directed acyclic graph with only
one "algebraic-exponential" activation on any path to the output (e.g., $\sin$
activation, $\tanh$ activation) and arbitrary piecewise-polynomial activations
otherwise (e.g., ReLU), cannot achieve the arbitrary finite approximation
property.
Results for networks achieving finite approximation but not being universal
formulas are slightly less general, due to technical challenges: the authors
show, for example, that classes $f(x) = c \sin (\omega \sigma(bx) + h)$ achieve
arbitrary finite approximation if and only if $\sigma$ is non-polynomial, and
demonstrate that for a certain sub-class of $\sigma$ maps studied earlier
involving $\sin$ activations, the resulting class does not have the universal
formula property. This kind of mapping can be viewed as a one-hidden-layer
neural network with one neuron on the hidden layer and $\sin$ activations; they
conjecture that the same separation holds for similar architectures with
multiple neurons in the hidden layer.
With these results established, they discuss known results on architectures
that do constitute universal formulas, and possible paths to characterize
necessary and sufficient conditions to be a universal formula further.



**Strengths:**

The writing, both technical and expository, is clear and precise. The authors
go to great lengths to present a broad view of the problem and prior work in
the introduction, which makes the subsequent technical analysis accessible to a
non-expert reader.

The authors defer more technical proofs to the appendices, but always give a
clear indication in the main submission what the types of arguments developed
are (e.g., algebraic methods vs.\ Pfaffian function theory). This will make the
work useful for follow-up work.

The discussion section (section 8) seems uncommonly interesting for a
submission in this venue, in that it lists several open problems that the
authors' work suggests, and which should lead to interesting follow-up work.


**Weaknesses:**

It is not a weakness per se, but as the authors themselves note, the
investigation is mostly of mathematical interest as "... the universality of
universal formulas is... an idealization that requires their parameters to have
unbounded precision or magnitude". Nonetheless, there are neural networks
commonly used that have structures similar to some of the near-universal
formulas studied (the authors mention SIREN, which uses periodic activations in
neural radiance field learning), and these kinds of periodic constructions have
been shown to be useful in studying general deep/shallow neural network
approximability (e.g., work of Telgarsky).



**Questions:**

### Questions and Comments

Does the "GeLU" activation function ($x \mapsto x \Phi(x)$, where $\Phi$ is the
standard normal cumulative distribution function) satisfy the
algebraic-exponential property? I am not sure if it would, because it is
defined with an integral, but if so, it might be interesting to mention here
(this activation is used in some modern transformer architectures).


### Minor Fixes

Caption of Figure 1: typo "uiversality"

Line 186: the $2\pi$ is on the wrong factor in the quotient? (the reals,
"modulo" $2\pi$).

Line 187: is this supposed to be the "fractional part" operator (based on its
specified domain) rather than the "floor" operator (which as I understand gives
the next-lowest integer)?

Line 345: "Then the existence of a ..." -- should this be "Then there exists
a..."?


**Limitations:**

Yes.

---

> ### Author Rebuttal · Authors · 2023-08-04
>
> Thank you for the careful reading of our paper and giving a very detailed and careful summary of our work.
>
> Indeed, the GeLU function unfortunately does not belong to our polynomial-exponential-algebraic class.
>
> All your minor fixes are spot-on (except for the last one: we did actually mean "existence" because our statement was about the decidability of an existence predicate; we will rephrase this sentence to make it clearer).
>
> Thanks again for many useful comments and positive evaluation of our work!

---

### Author Rebuttal · Authors · 2023-08-05

We thank all the reviewers for the careful reading of our work and useful feedback, both positive and critical.

As we mention in the response to Reviewer 2cHj, our work is partly motivated by an experimental observation that neural networks with the activation function $\sin$ apparently can easily overfit. We attach a .pdf figure illustrating this with a simple example of a two-hidden-layer network fitted by gradient descent to a one-dimensional target function. The network fits the training set reasonably well, but wildly oscillates outside. This is the sort of behavior we expect from models in our class $\mathcal G_1\setminus\mathcal G_2$ (trainable, but poorly generalizing). A ReLU-networks of the same architecture is trained to a comparable accuracy on the training set, but shows a much better generalization. In a weaker form, we also observe this effect in MNIST classification: $\sin$-networks again seem to be somewhat more prone to overfitting than ReLU-networks of the same architecture. Our theoretical setting of universal formulas can be viewed as an extreme case of these experiemental setups, when the network size is small and fixed, the weights are trained by some ideal optimization algorithm, etc. The above experimental observations seem to agree with our theoretical results and conjectures regarding the multi-layer $\sin$-networks being in the class $\mathcal G_1\setminus\mathcal G_2$, in contrast to ReLU-networks.

---

> ### Comment · Reviewer_FCbX · 2023-08-11
> **Unconvinced about intuition**
>
> The behaviour of the neural networks with sine activation function is more closely related to the "deep Fourier expansion", see Theorem 6.1 of [1], and to super-expressive activation functions, see [2].  This is not a qualitative phenomenon, but a quantitative phenomenon, rooted in the fact that any model class which approximates at a sufficiently fast rate, must be unstable, see [3].
>
> Concerning generalization, please see point 1 in my response to your rebutal.  There I explicitly show, using a basic dominated convergence argument, that models which cannot uniformly approximated but can approximate pointwise can asymptotically achieve 0 risk.  Therefore, I cannot see why models in $G_1 --G_2$ are expected to generalize poorly; mathematically.  Since this is Mathematical paper I expect the authors to substantiate this point theoretically.
>
> [1] Yarotsky, Dmitry, and Anton Zhevnerchuk. "The phase diagram of approximation rates for deep neural networks." Advances in neural information processing systems 33 (2020): 13005-13015.
>
> [2] Yarotsky, Dmitry. "Elementary superexpressive activations." International Conference on Machine Learning. PMLR, 2021.
>
> [3] Petrova, Guergana, and Przemysław Wojtaszczyk. "Lipschitz widths." Constructive Approximation 57.2 (2023): 759-805.

---

### Decision · Program_Chairs · 2023-09-21

**Decision:**

Accept (poster)

**Comment:**

The paper explores the expressive power of "universal formulas" which are fixed-complexity function families on $[0, 1]$. This is in contrast to "universal approximation" results for neural networks that state that neural networks can approximate any continuous function on $[0,1]$ by increasing the width sufficiently (changing the complexity). The authors present a hierarchy of three nested expressiveness classes and show interesting separations where a function class may be able to fit any finite size set but no generalize to the rest of the domain. They connect their observations to explain overfitting by $\sin$ activations. The contributions are summarized exceptionally well by [reviewer BXX4](https://openreview.net/forum?id=gmVoaAxB1R&noteId=5DqlHjhgWB).

The main drawback of the work is perhaps that it is mostly a mathematical exploration given the use of infinite precision, and one dimensional functions. Nevertheless, as all the reviewers agree, the results are interesting and novel, and useful for the community to know. Therefore, I recommend acceptance. I encourage the authors to incorporate the feedback from the reviewers (as they see fit), and add discussions about connections to approximation theory.